# Analysis of Sustainable Public Transportation and Mobility Recommendations for Montevideo and Parque Rodó Neighborhood

**Silvina Hipogrosso** [1,*,†] and **Sergio Nesmachnow** [1,2,†]

1   Universidad de la República, Montevideo 11300, Uruguay
2   South Ural State University, 454080 Chelyabinsk, Russia; sergion@fing.edu.uy
*   Correspondence: silvina.hipogrosso@fing.edu.uy
†   These authors contributed equally to this work.

**Abstract:** This article presents an analysis and characterization of recent sustainable initiatives developed for the public transportation of Montevideo, Uruguay. In addition, specific analysis and recommendations are proposed for the Parque Rodó neighborhood, based on a survey performed to people that commute to/from that area. The analysis considers the main concepts from related works, evaluating relevant quantitative (coverage, accessibility, affordability, etc.) and qualitative (public finance, integration, comfort and pleasure, etc.) indicators. Three sustainable public transportation initiatives are studied: electric bus, public bicycles, and electric scooters. Results of the analysis for each transportation mode suggest that the first initiatives focus on specific sectors of the population and should be improved in order to extend their accessibility and affordability. In turn, coverage must also be expanded. Regarding the analysis of the Parque Rodó neighborhood, results indicate that people are willing to perform the modal shift to more sustainable transportation modes, but several improvements are needed to improve the quality of service. All these aspects are considered in the proposed guidelines for a sustainable mobility plan in the area and also for suggestions and recommendations formulated to develop and improve sustainable mobility in Montevideo.

**Keywords:** sustainable mobility; public transportation; smart cities; mobility plan

---

## 1. Introduction

Mobility is a crucial component of modern society, where the participation of citizens in social, economic, and cultural activities requires traveling over both short and long distances [1]. Sometimes, traveling takes citizens a long period of time, regardless of the distance traveled, due to many reasons related to mobility situations. The ability of individuals to overcome limitations imposed by time, distance, and other mobility-related difficulties is critical to guarantee an active participation in city life [2].

Sustainable mobility is a subject that studies the development and use of transportation modes that are sustainable regarding several matters, mostly economic, environmental, and social [3]. Assessing sustainability and studying alternative transportation modes is very important considering that transportation largely contributes to environmental pollution with direct negative implications in health and quality of life of citizens. This is a relevant subject of study under the novel paradigm of smart cities [4].

Modern urban areas have been built around automobiles. This transportation mode has dominated the urban landscape, gaining the majority of the space on streets, limiting pedestrians zones, and reducing the space for other (sustainable) transportation modes. Private vehicles have

revolutionized mobility, but they have also introduced several problems, including pollution, environmental degradation, congestion, accidents, a decline in public transportation, lack of accessibility, etc. However, nowadays many cities across the world are designing sustainable mobility plans to limit the use of private cars by improving public transportation, encouraging sustainable transportation modes, creating pedestrian zones, and changing the negative transformation caused by automobile dominance. While acknowledging the importance of car mobility in modern cities, this article focuses on analyzing sustainable transportation initiatives as an important mean of promoting the shift from cars to more sustainable transportation modes.

One of the most sustainable modes for mobility is provided by public transportation systems [5]. Public transportation allows moving a significantly larger number of people than private transportation, using a fewer number of vehicles. Furthermore, public transportation contributes to reduce greenhouse gas emissions, significantly improving the pollutant per passenger/km when compared with private vehicles. In the specific case of Montevideo, Uruguay, just a few initiatives have been recently proposed to promote sustainable private mobility (e.g., electric taxis and electric vans for last mile distribution of people and goods [6]). On the other hand, several recent initiatives have been proposed for sustainable public transportation, which are under development.

In this line of work, this article studies sustainable mobility initiatives recently developed in Montevideo, Uruguay: electric bus, public bicycles, and electric scooters. The main motivation of the study is to analyze and characterize the current reality regarding sustainable public transportation in Montevideo, in order to make mobility more sustainable. In turn, a specific analysis is performed on Parque Rodó neighborhood and Engineering Faculty, based on a survey performed to 617 citizens that travel from/to the studied area. Sustainable mobility alternatives are reviewed and categorized, and the main opinions about sustainable transportation modes are summarized and analyzed. Specific suggestions and recommendations are provided to develop and improve sustainable mobility in Montevideo and in Parque Rodó neighborhood.

This article extends our previous conference presentation "Sustainable mobility in the public transportation of Montevideo, Uruguay" [7] at II Ibero-American Congress on Smart Cities. New content and contributions in this article include (i) an expanded review of the related literature about sustainable mobility and proposals for Montevideo; (ii) the analysis of current sustainable initiatives in the public transportation system of Montevideo, regarding quantitative and qualitative indicators; (iii) suggestions and recommendations to develop and improve sustainable mobility in Montevideo; (iv) the analysis of the current situation regarding transportation and sustainable mobility in Parque Rodó neighborhood and Engineering Faculty; and (v) suggestions and recommendations proposed to develop and improve sustainable mobility in Parque Rodó neighborhood.

The article is structured as follows. Section 2 presents the main concepts related to sustainable mobility. A review of the main related work is presented in Section 3. The analysis of current sustainable mobility initiatives in Montevideo and specific recommendations are reported in Section 4. The analysis of the Parque Rodó neighborhood and Engineering Faculty, and the suggestions for developing and improving sustainable mobility in that area, are described in Section 5. Finally, Section 6 presents the conclusions and the main lines of future work.

## 2. Sustainable Mobility

Sustainability has been a major concern of modern society since the last decades of the twentieth century. Furthermore, sustainability has become a crucial aspect for nowadays communities, due to its direct implications on quality of human life and the growing awareness of the main issues and threats posed by environmental problems.

In 1987, the Brundtland Report for the World Commission on Environment and Development introduced the term sustainable development, to define "the development that meets the needs of the present without compromising the ability of future generations to meet their own needs" [8]. Sustainable development has become a paramount rule for modern sustainable mobility, i.e.,

the process to guarantee the movement of people with minimal environmental impact. Indeed, the World Business Council for Sustainable Development defined sustainable mobility as the ability of a society to fulfill requirements related to the movement of people without sacrificing fundamental human or ecological values [9]. Sustainable development is studied through three interrelated areas: the social, environmental, and economic dimensions. In several documents from the United Nations, sustainable development is said to be achieved when the goals of social equity, viable economic, and environmental friendliness are met in a coordinated manner [10]. Transportation and mobility are keys for sustainable development. Sustainable transportation can improve economic growth as well as improve accessibility (a very relevant social issue). Sustainable and safe transportation achieves a better integration of the economy while respecting the environment. As a consequence, sustainable mobility solutions must be designed considering these three areas, depicted in Figure 1, to contribute positively to their communities.

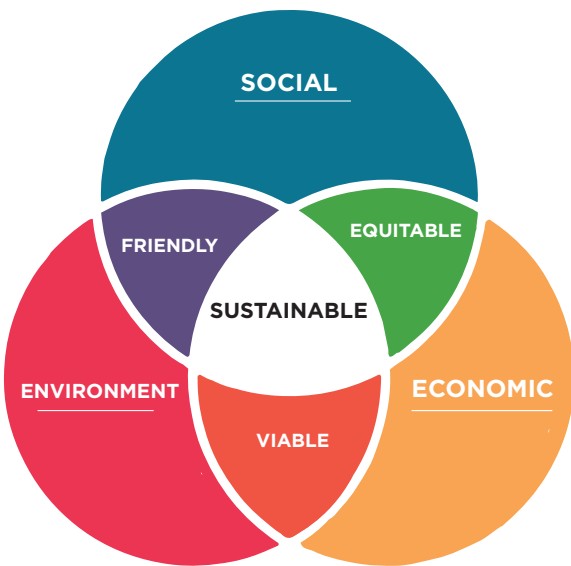

**Figure 1.** Main areas related to sustainable mobility.

The sustainable mobility paradigm integrates many relevant concepts [11]. Among them, a new approach has been proposed for designing and planning transportation systems, based on social processes, accessibility, reduction of motorized transportation, integration of people and traffic, and other factors to consider mobility as a valued activity, regarding environmental and social concerns [12].

Traditional research on sustainable mobility has focused on environmental impacts. However, several other important aspects have also been analyzed recently, including the relation with equity and the impact on economy, safety, health, and quality of life in general. In this regard, technology has been identified as one of the main tools that helps ensuring energy efficiency, using alternative and renewable energy sources, reducing contamination (e.g., pollutants emissions, noise, etc.), guaranteeing environmental friendliness, and providing tools for the analysis of reality and the development of new approaches based on data analysis. Furthermore, developing a rational plan for economic investments (e.g., infrastructure, transportation modes, etc.) is crucial to promote sustainable and solid growth in the long-term. Modern plans or courses of action are oriented to bring cities onto a more sustainable path with special emphasis on sustainable mobility, raising awareness, and involving citizens, with the main goal of fostering a behavioral change. The ultimate purpose is that citizens realize that transportation modes proposed by the sustainable mobility paradigm helps society, thus they opt for using more sustainable options by their own.

Many cities around the world aim at creating more compact, well-connected, and coordinated communities to reverse the development trend that focuses on private car and provides poor access to public transportation. Among several proposals for urban planning, Transit Oriented Development (TOD) stands as a model capable to contribute towards long-term environmental sustainability, in opposition to unrestricted growth usually happening in urban areas [13]. TOD promotes developing dense, compact neighborhoods that comprehend many uses (residential, commercial, cultural, institutional, etc.) physically connected by pedestrian and bicycle friendly transportation, also efficiently integrated with mass transit systems to connect with other zones within a city. The compact and integrated city design proposed by TOD also helps reducing carbon emissions, by providing people access to services at a walking distance. TOD also promotes socioeconomic benefits, expanding the connectivity of public transportation with housing and economic opportunities, which in turn enables developing more socially equitable cities by improving access to job opportunities and services.

Several indicators have been proposed and developed to study sustainable mobility in urban scenarios [14]. Among them, the most relevant are commuting travel time, coverage, access to mobility service, affordability, comfort and pleasure, intermodal connection, and integration. Indicators have been applied to analyze different transportation modes in many cities around the world. Some of the main related works on the topic are reviewed on next section.

## 3. Related Work

Modern cities are conceived to support different transportation modes, which are supposed to coexist, interoperate, and share the urban space. To guarantee a proper support for intermodality, transportation modes are supposed to be well integrated and connected, thus providing citizens with efficient and effective mobility [15]. However, the transportation planning problem is usually driven by traditional approaches that simplify, or even ignore, the complexity of handling several transportation modes coordinately. As a consequence, administrators often fail to provide holistic plans, accounting for all transportation modes operating in the city and including a comprehensive decision-making to consider indirect and interrelated impacts of the implemented solutions. Approaches that do not consider the city and transportation systems as a whole lead to isolated actions that usually result in poor and inefficient policies, which fails to solve the main problems related to mobility, among them, sustainability.

Litman and Burwell [16] established the relationships between sustainable transportation and mobility and recognized that in order to achieve sustainability, transportation must be conceived from a broad point of view to consider energy efficiency, health, economic and social welfare, and other relevant aspects related to sustainable development. Transportation impacts on sustainability were characterized into three broad areas (economic, social, and environmental) and it was stated that the correct approach for solving the underlying issues is to find strategies that help achieving all the main goals (in the long-term) by increasing transportation system efficiency. Several perspectives for addressing the sustainability problem in transportation were reviewed, and a list of common indicators for sustainable transportation was presented. Approaches for sustainability in transportation include improved travel choices, pricing and road design incentives, patterns for land utilization, and technical improvements to motorized vehicles, among others. A paradigm shift [17] was proposed for rethinking transportation, to consider different integrated solutions to achieve sustainable transportation systems.

Sustainable urban mobility planning begins with designing a strategic plan for the community. Banister [11] put special emphasis on stakeholder participation in the planning process, to involve them in the reasoning and implementation of specific initiatives for sustainable mobility. After questioning conventional principles for transportation planning, seven key elements for sustainable mobility were discussed and a more flexible paradigm to meet sustainable mobility purposes was proposed. Banister concluded that proactive involvement of relevant actors (including academia, policy-makers, and others) is crucial and more effective than traditional approaches for sustainable mobility. In this regard, specific changes must be made regarding land utilization, environment, public health,

and ecology. Actors must accept their collective responsibility to achieve an effective sustainable mobility model to overcome car dependency. In particular, stakeholders must be engaged to support the application of measures towards promoting the main goals of sustainable mobility.

Methodological analysis and indicators have been studied as useful tools for the evaluation of sustainable mobility in cities [14,18,19]. Relevant studies are those related to understand the evolution of current transportation systems towards sustainable one, and those that evaluate the impact of selected solutions for specific case studies. In this context, indicators are used to simplify complex phenomena and to provide hints of different issues or problematic situations [20]. The combination of multiple indicators also allows capturing different dimensions and aspects of sustainable mobility.

The main concepts about indicators for sustainable transportation were presented by Gudmundsson et al. [14], focusing on the role and importance of quantitative and qualitative indicators for stakeholders (including decision-makers, planners, and operators). Frameworks for assessing sustainability metrics were reviewed, and a framework towards sustainable transportation was proposed. Two case studies were presented: European transportation and high-speed rail in England. The main conclusions from the case studies is that indicators and frameworks strongly depends on the context. Thus, understanding the context is key for succeeding in implementing effective sustainable mobility actions. Solid frameworks can stimulate politicians, decision-makers, and citizens to cooperate in the implementation of practical efforts regarding the studied topic.

Miller et al. [5] studied the role of public transportation regarding sustainability, reviewing related works that analyzed case studies of sustainable transportation. Frameworks, key challenges, and the benefits of public transportation were analyzed considering the three dimensions from sustainable development (environment, economy, and society). The reviewed articles were characterized regarding the main considerations for each of the studied dimensions. Finally, a set of recommendations were provided for developing and planning sustainable public transportation systems.

Rodrigues et al. [21] developed an index of sustainable urban mobility including several important features previously identified by Litman [17] for comprehensive and sustainable transport planning. The index is based on data obtained from planners and includes weights for different criteria, defined by experts. An application was presented for the city of São Carlos, Brazil, a medium-size city with 250,000 inhabitants. A basic analysis was proposed to demonstrate the viability and the efficacy of the proposed index. The main results of the study indicated that the proposed index was relatively easy to compute and flexible enough to be applied to characterize sustainable mobility.

Johnston [19] developed Production, Exchange, and Consumption (PECAS), a comprehensive method for modeling the impacts of transportation, to be included in an integrated urban model of California. PECAS was conceived as an spatial economic urban model, combining Walrasian concepts and random utility theory, using a network and a zone-based travel model to provide a theoretically valid measure of regional and statewide utilities. PECAS provides several data of economic utility like households by income, housing rents, housing affordability, etc. Several major transportation scenarios were studied, including metropolitan planning organizations of California, Sacramento, and San Diego. The metric evaluated included greenhouse gas emissions, air pollution, and economic welfare and equity. Results of the analysis were reported as relevant to state and regional transportation plans.

Baidan [22] studied the main problems of public transportation in Bucharest, Romania, which are similar to other capitals in Eastern Europe (e.g., the lack of policies to discourage car use, the lack of intermodal transportation options, etc.). Public transportation in Bucharest was analyzed through an accessibility indicator, considering the fares and the access of new residential areas to the transportation system. The main results of the study indicated that Bucharest has the cheapest public transportation fares among post-socialist European capitals, and that even though the metro does not cover many of the new residential areas, most of the citizens living there do have access to surface public transportation. A relevant suggestion was formulated, proposing the creation of intermodal hubs; networking the local/regional train systems with the metro; and providing other services as Park&Ride, bicycle parkings, rental services, etc., to promote sustainable transportation to citizens.

The successful case of sustainable mobility in Bogotá, Colombia, was analyzed by Lyons [23]. The study focused on specific actions oriented to address the protection of the environment and achieve both economic and social sustainability via a non automobile-centric approach. To satisfy the mobility needs, the city administration developed a mobility strategy based on three pillars: (i) discouraging the use of the car; (ii) the promotion of bicycle and walking as transportation modes; and (iii) the construction of an efficient public transportation network, anchored by a Bus Rapid Transit (BRT) system. Bogotá integrated transportation planning with social planning, by designing open areas and housing plans accessible to public transportation. According to Lyons, the results of the actions made in Bogotá span the three pillars of sustainability: economic, social, and environmental. Some of the outstanding results for the case study of Bogotá include BRT reduced travel times in 32%, 9% of former car drivers switched to BRT, traffic accidents reduced 93%, and air pollution decreased 40%. Overall, the study argues that the Bogotá case demonstrated that BRT is a viable and realistic solution as urban transportation plan, and also that increasing infrastructure for bicycle and walking is a viable strategy to promote sustainable transportation modes. The author concluded that the case study of Bogotá can be replicated in other developing countries, on the path towards sustainable transportation.

Moreover, in Latin-America, Rodrigues et al. [18] studied the provision of transportation services in Brazil following a specific methodology based on workshops with public managers and planners to characterize sustainable urban mobility. Multiple criteria analysis was applied to give support for decision-making, in order to identify proper goals, evaluate their relevance, and assess the impact of different solutions. Four stages were applied: (i) characterization of the problem; (ii) identification of relevant elements (including accessibility, congestion, infrastructure, social inclusion, pollution, non-motorized modes, and others); (iii) construction of a cognitive map, using operational and strategic concepts to reach the objectives; and (iv) identification of key viewpoints, using relevant concepts from the cognitive map, according to decision-makers. A series of workshops were performed in eleven Brazilian cities to gather information. Results were grouped by geographical regions, focusing on capturing different dimensions of sustainability in the context of each region, regarding three dimensions: social, economic, and environmental. The main identified problems were related to the relevance of urban public transportation, infrastructure, and environment. The importance of social, economic, and environmental issues reflected the development of each studied region. In conclusion, Rodrigues et al. stated the importance of the applied methodology to capture different views of sustainable mobility in Brazil and its application for creating public policies in that regard.

In Uruguay, project URU/17/G32 "Towards a sustainable and efficient urban mobility system in Uruguay" was launched in 2017, as a joint effort of government and transportation companies. The main goals of the project are defining regulations for low-carbon transportation systems, evaluating clean technologies in Montevideo, and promoting a cultural change towards sustainable transportation modes. Other recent initiatives for studying and developing sustainable transportation in Montevideo are project MOVES [6], which aims at promoting an effective transition towards inclusive, efficient, and low-carbon urban mobility in Uruguay, and project "Public transportation planning in smart cities" [24], funded by Fondo Conjunto de Cooperación Uruguay–México (2018–2019).

Regarding sustainable mobility plans for neighborhoods or specific zones of Montevideo, the case of Parque Rodó neighborhood presented in this article is the first study proposed so far. Current projects URU/17/G32 and MOVES have developed some initiatives regarding this subject, but focused on small and isolated cases, such as the Institutional Plan for Sustainable Mobility developed at the Uruguayan office of the United Nations Development Programme, with the goal of motivating people to adopt healthy mobility habits. This initiative is reported as "started", but no details about the methodology or results have been published yet. In any case, this is a very low-impact initiative, involving less than 20 persons that commute to the UNDP workplace [25]. Other minor initiatives related to sustainable mobility have been developed, but without proposing or implementing a sustainable mobility plan, such as replacing (non-sustainable) delivery vehicles for electric tricycles and electric pedal-assisted bicycles and incorporating electric utility vehicles for cargo companies.

Current project "Spatial, universal, and sustainable accessibility: characterizing the multimodal transport system of Montevideo, Uruguay" (code FSDA_1_2018_1_154502), funded by Fund for Research from Open Data, National Agency for Research and Innovation, Uruguay, proposes developing a characterization of urban accessibility, as an important tool to determine the quality and equity of transportation systems and the impact on daily activities of citizens. Several data sources are studied to characterize the accessibility of the transportation system of Montevideo. Three dimensions are considered in the analysis: territorial, universal, and sustainable accessibility. Territorial accessibility analyzes existing mobility alternatives to identify potential accessibility problems that prevent the participation of citizens in social and economic activities. The universal dimension studies accessibility problems of transportation systems for people with reduced mobility and the elderly, the alternatives of universally accessible transportation modes, and the identification of points of interest that are not universally accessible. Finally, regarding the sustainable dimension, sustainable mobility options offered by the city are studied, including non-polluting and low emission vehicles. A fourth dimension is proposed for the study, transversal to the previous three, to assess the impact of the socioeconomic condition on the three accessibility dimensions studied. It is expected that results from the project will be useful for the authorities, transportation companies, academia, and civil society in general, to identify situations of inequality in access to the different modes of transportation in the city.

The research reported in this article is a novel proposal for Uruguay, oriented to evaluating current sustainable mobility initiatives in Montevideo and Parque Rodó neighborhood. The research is part of current project "Spatial, universal, and sustainable accessibility: characterizing the multimodal transport system of Montevideo, Uruguay", developed within the sustainable mobility dimension.

## 4. Sustainable Public Transportation Initiatives in Montevideo

This section describes and analyzes sustainable mobility initiatives that are operating in Montevideo through public or private transportation companies.

### 4.1. The Public Transportation System of Montevideo, Uruguay

Public transportation in Montevideo is mainly comprised of buses. Taxis and other minor systems also operates in the city. Regarding the bus system, city authorities proposed the Metropolitan Transportation System (Sistema de Transporte Metropolitano (STM)), an urban mobility plan with the main goal of restructuring and modernizing public transportation in Montevideo [26]. Public transportation was integrated into a unified system comprised of 1528 buses operated by four private companies. The bus network consists of 145 bus lines, but considering the different variants of each lines, the number increases to 1383, a remarkably large number for a city like Montevideo.

The city center is a hub in the bus network, with most lines converging to that area. Additionally, the large length of certain bus lines with respect to the area of Montevideo is also noteworthy: the average median bus line length is 16.4 km, with the longest line spreading over 39.6 km, also a large number, as the area of Montevideo is 530 km$^2$.

In 2019, taxis were integrated in STM. Passengers can pay the rides on any transportation modes using the STM smart card, contactless top-up cards that are linked to the identity of the owner. STM can also used to pay for rides in the public bicycle system, as described in the next section.

### 4.2. Analysis of Sustainable Mobility Initiatives in Montevideo

Several sustainable mobility initiatives have been developed in the last years in Montevideo, including a prototype of an electric bus system (operating with just one line), a system for shared public bicycles, and a system for on-demand mobility using electric scooters. Other initiatives and measures to promote sustainable mobility and discourage the use of private motorized vehicles have been developed. The main details of each initiative are presented in the following subsections.

### 4.2.1. Electric Bus (Pilot Plan)

As in most countries in Latin America, Uruguay has recently prioritized moving towards cleaner energies in public transportation, in order to reduce its carbon footprint. For the last few years, transportation authorities in Uruguay have studied the potential benefits of including electric vehicles to the public transportation fleet in Montevideo. As a result, the Uruguayan government requested a loan from the Green Climate Fund, the entity that operates financial mechanisms to assist developing countries in adaptation and mitigation practices to counter climate change, in order to facilitate the modal shift from diesel to electric buses and allow Montevideo to replace 10% of the bus fleet [27].

The main public transportation company operating in Montevideo is CUTCSA, accounting for about two-thirds of the market share and also of the buses operating in the city [28]. CUTCSA has conducted tests of mobility using electric buses, with incentives and support from the Ministry of Energy and the city administration of Montevideo. Since 2017, a pilot plan is in course, using one electric bus that operates in different lines (the line changes weekly) to test the performance of this new transportation mode. The electric bus used in the pilot plan is a fully electric (no emissions) ByD vehicle, model K9A, with an autonomy of more than 250 km. It has an environmental friendly long-life iron phosphate battery of 324 kWh that admits more than 6000 charge cycles, and demands 3.5 h for total fast charge. The maximum speed of the bus is over 90 km/h and the average consumption is 100 kWh each 100 km. The pilot plan using electric bus has been considered an important advance for public transportation system of Montevideo. The new buses incorporates air conditioning to keep the environment ventilated, has decreased motor sounds and vibration, and offers universal accessibility, which is a great improvement in particular to those people with reduced mobility.

In 2019, 30 electric buses circulated on Montevideo with the purpose of gradually evaluating their operation and integration to the existing public transportation system. During 2020, CUTCSA plans incorporating 120 new electric buses to the transportation system of Montevideo [29].

### 4.2.2. Public Bicycles

The public bicycle system implemented by the city administration of Montevideo in 2015 is called Movete. It was conceived as part of the urban transportation system, in order to promote green mobility and a healthy way to know the city, move to workplaces and other relevant individual locations, and also to extend the accessibility of the public transportation system to specific final destinations.

As of December 2019, the public bicycle system consists of a fleet of 80 bicycles spread in a network of eight automated stations, distributed from the Old City to the Center neighborhoods. Bicycles can be rented at one station and returned in another station in the area covered by the system. A card of the integrated Metropolitan Transportation System (STM) is required to rent a bicycle. People that do not own a STM card, e.g., tourists or new users, can obtain it with no charge in the center office of Movete. The service has a time limitation for users, which cannot use the bicycles for more than four hours per day. This limitation is in line to promote more people have access to use the service.

The city administration is planning to expand the coverage (i.e., the area of operation) of the public bicycle service in 2020. The expansion is planned to include 60 stations and 600 bicycles, in order to increase accessibility and promote active mobility. New neighborhoods near the city center will be covered by the service, including Cordón, Parque Rodó, Parque Batlle, and Tres Cruces, accounting for a significant larger population then in the current implementation of Movete.

### 4.2.3. Electric Scooters

The electric scooter is a new mode of urban transportation that has gained popularity all over the world as an alternative to driving. Electric scooters provide an environmentally friendly alternative for short journeys that are either too far to walk, or too close to drive a car, to be a cost-effective option. Three companies of electric scooter (Grin, Lime, and Movo) operated in Montevideo since 2018, but two of them (Movo and Lime) stopped operating on December 2019.

Grin operates with an application where electric scooters and electric scooter stations are shown in a street map of Montevideo. The electric scooter stations are parking places where the company park scooters, with the permission of a close local business or an institution. Stations also provide a connection to the electric grid to charge scooters.

The service provides a practical and easy way to use electric scooters: by simply downloading a mobile application and setting up a payment method, users have access to a network of scooters that they can use at any time. Electric scooters have GPS blue tracking, so users are never too far from picking up a electric scooter and they can leave it anywhere within the area where the service operates.

To reduce logistic efforts of collection and distribution, Grin incorporated scooter stations after negotiations with drug stores, education centers, parking lots, and other commercial businesses around the city. Furthermore, the stations allow charging scooters. However, people still park scooters in any place. Up to December 2019, the company did not apply any penalty fee for not using the stations.

### 4.2.4. Other Initiatives

Other initiatives for sustainable mobility have been developed in Montevideo recently. There exist brand-new services that are not fully operative at the time of writing this article (February, 2020), or no data is available to perform a sound evaluation and analysis. Two of the main new initiatives are electric taxis and hybrid car sharing.

Electric taxis. The electric taxis initiative is in line with the main idea of government entities to promote using energy efficiency in public transportation, which have promoted the shift to more renewable energy sources, especially since 2015. As the change from gasoline to electric implies a higher initial investment, it is more profitable on taxis, which run many kilometers and the investment can be compensated with the lower operating cost. Currently, 54 taxis circulate on the streets of Montevideo and the main goal of the city administration is to increase the fleet of electric taxi to 300 vehicles (10% of the total fleet of taxis) by the end of 2020 [30].

Hybrid car sharing. The hybrid car sharing service provides citizens a mean to rent cars on-demand, for short periods of time. The system is accessed through a mobile application that allows users to choose from different locations to pick up and return the car, and the time they will use the service. The car sharing facilitator is a car brand (e.g., Toyota is one of the car brands providing this service in Montevideo). To promote the use of the hybrid car sharing service, the Uruguayan government has exonerated parking costs in the city for this transportation mode. Carsharing is a very new service in Montevideo (operating since late 2019), so there is not enough information or data to perform an in-depth analysis.

### 4.2.5. Promotion of Walking and Discourage the Use of Private Motorized Vehicles

Besides the aforementioned sustainable public transportation and sustainable mobility initiatives, other actions have been performed by the city administration to promote pedestrianism and reduce the number of private motorized vehicles in circulation in Montevideo.

Pedestrianism. Walking is the most sustainable transportation mode because is the only one that does not depend on any device or service. In Montevideo, multiple initiatives have promoted and stimulated pedestrianism; among the most relevant ones we can mention that walking lanes were incorporated in several parks, several streets in different neighborhoods (Old City, Reus, and even in low-income peripheral suburbs) were transformed for pedestrian-only use, and the constant reparation of sidewalks in a joint initiative (the Sidewalks Plan) with the participation of the municipality and residents. A relevant project to improve pedestrianism, which involved several infrastructure modifications, is the "Old city at human scale" project. Focused on the Old City neighborhood, the aim of the project is revitalizing the historic center of Montevideo, promoting sustainable mobility, universal accessibility, and improvement of public space [31]. Several important tasks were developed within the project, including repairing and widening all sidewalks, transforming them to single pavement; incorporating access ramps in every corner to improve accessibility; renovating the public lighting to

improve safety; highlighting historical buildings and renewing urban equipment (street and square benches, litter bins, gardening, and signs, which were unified to give more information to the user); and building new rest areas, as suggested by neighbors. The project involved other activities related to the improvement of tourism service, environmental management, renovation of buildings that were badly damaged, and the incorporation of bicycle lanes all around the historic center. The project achieved positive results, thus improving accessibility, comfort on walking, safety, urban equipment on public spaces, and sustainability of the area.

Measure to discourage the use of private motorized vehicles. There is a rising concern about the impact of automobiles in the urban mobility area, which cause air pollution, human health effects, global climate change, congestion, and noise pollution. In order to achieve sustainable mobility, it is not enough to promote sustainable transportation modes; specific policies to limit the use of private motorized vehicles must be applied too. In this regard, the city of Montevideo has also proposed several measures to discourage the use of cars, thus contributing to sustainable mobility. Some policy measures applied include bus-only and preferential lanes for buses and taxis were deployed in many avenues, to promote the use of public transportation; tariff zones were applied for street parking in districts with high traffic and congestion (Old City, Downtown, and Cordón neighborhoods), the development of new sustainable transportation modes has been considered (e.g., an electric train connecting the city center with the Eastern part of the city) to reduce the dependence on automobile, the integrated public transportation system of Montevideo has recently incorporated taxis. All these measures are in line with the efforts to reduce car utilization. Furthermore, the administration has recognized the importance of promoting a cultural change, for citizens to do a responsible car utilization. Furthermore, project "Old city at human scale" also discourages the use of the automobile, taking away street space to give it to urban equipment and pedestrianization.

Other relevant measures, such as private traffic restrictions, which play a significant role in urban transportation regarding accessibility, air quality, and other factors that affect the quality of life of citizens [32,33], have not yet proposed nor implemented in Montevideo.

### 4.3. Indicators to Assess Sustainable Mobility in Montevideo

The proposed analysis considers a subset of sustainable mobility indicators proposed by the World Business Council for Sustainable Development [9]. The analysis applies a mixed approach, considering quantitative and qualitative indicators, which is the dominant methodology for sustainable mobility analysis, according to the review by Anagnostopoulou et al. [34]. On the one hand, quantitative indicators are those for which the available data for the case study allows computing a numerical value to assess a sustainable mobility criteria. On the other hand, qualitative indicators are metrics based on opinions, feelings, or points of view, rather than specific facts or numbers. Qualitative analysis are applied when the relevant pieces of information are not available for the studied initiatives. In particular, for the case of study of Parque Rodó neighborhood, a survey based on specific interviews to a sample of 617 persons moving to/from the studied area was performed. Resulting data were analyzed both using quantitative and qualitative indicators.

#### 4.3.1. Quantitative Indicators

The quantitative indicator group includes coverage, access to mobility service, affordability, and commuting travel time. The corresponding definitions are presented next.

Coverage. The coverage is defined as the ratio of the area covered by each sustainable mobility service ($ci$) and the total urbanized area of the city ($ta$), according to Equation (1). The total urbanized area of Montevideo is considered to extend for $200\,km^2$. The scale for this indicator is straightforward, 0 correspond to 0% of coverage and 10 correspond to 100% of coverage.

$$coverage = \frac{ci}{ta} \tag{1}$$

Access to mobility service. Access to mobility service (*am*) is defined as the share of population with appropriate access to each service, according to Equation (2), where *nh* is the number of citizens living in the city, and *PR(i)* is the percentage of people living within 400 m from a public transportation stop or from a possible renting point of a shared mobility system.

$$am = \frac{\sum_i PR(i)}{nh} = 1 - \frac{\overline{PR}}{nh} \qquad (2)$$

The methodology for calculation implies determining the percentage of people living within the service areas by using spatial data analysis. The service area is limited by a distance of 400 m of a sustainable mobility service, which is considered as the maximum distance that a person considers to walk to use a public transportation service [35]. The scale for the *am* indicator is 0 represents 0% of the population in the city and 10 represents 100% of the population.

*Affordability of sustainable mobility transportation.* Affordability (*af*) is defined as the expenditure on transportation made by persons as a percentage of their income. The calculation is based on the methodology by Carruthers et al. [36], considering the cost of performing 45 and 60 trips on each transportation mode and on existing socioeconomic data. The indicator is computed for two different relevant social groups, considering the minimum income and the middle income per capita, according to values reported for 2019 by National Institute of Statistics, Uruguay [37]. The calculation method is described by Equation (3), where *nt* is the number of trips, *p* is the cost of a single trip, and *is* is the income per capita. The scale for the *af* indicator is 0 indicates affordability index is over 35% and 10 indicates that is less than 3.5%

$$af = \frac{nt \times p}{is} \qquad (3)$$

Commuting travel time. This indicator is defined as the average time spent by a person when travelling from origin to destination of a trip performed in the public transportation system. The methodology applied for calculation considers that (i) (for bus) the average commuting travel time includes the time for a person to walk to the bus stop and the time waiting for the bus to arrive; persons are supposed to walk from the centroid of the zone and the average walking speed is assumed to be 5 km/h; (ii) (for bicycles) the average speed is 13.5 km/h and the average walking time to a bicycle station is 4 min (walking up to 400 m); and (iii) (for scooters) the average speed is 12 km/h and the walking time a person takes to find a scooter is less than 3 min.

Commuting travel times are computed for two relevant distances: (i) a short travel of 3 km, a reasonable distance for travels to nearby locations such as offices, shopping, education, etc. It is also the average travel distance for electric scooters, considering an average speed of 12 km/h. (ii) A medium distance of 10 km, a reasonable average distance for travels to work, according to data from the urban mobility survey for Montevideo [38]. It is also the average travel distance on public transportation, considering an average bus speed of 13 km/h [39].

Two scales are considered for this indicator, for 3 and 10 km. Both consider as lower limit the time to travel the corresponding distance at the average human walking speed of 5 km/h, and as upper limit the time to travel the corresponding distance at the limit speed of bicycles and electric scooters (25 km/h). Thus, for the 3 km distance, 0 represent a trip duration of over 36 min and 10 represents a trip duration of 7 min, and for the 10 km distance, 0 represent a trip duration of over 2 h and 10 represents a trip duration of 24 min.

### 4.3.2. Qualitative Indicators

The qualitative indicator group includes net public finance, energy efficiency, intermodal connectivity, intermodal integration, and comfort and pleasure. The corresponding definitions are presented next.

Net public finance: Percentage of the cost of each mobility service that the government grants as subsidy to transportation companies.

Energy efficiency: Energy consumption in public transportation, usually evaluated in oil equivalent. The efficiency indicator considers the total energy demand from clean (i.e., renewable) and non-renewable sources.

Intermodal connectivity: Number of locations where users can change from one transportation mode to another.

Intermodal integration: Quality of the intermodal facilities between the different transport modes.

Comfort and pleasure: Satisfaction perceived by citizens about comfort and pleasure of moving in the city using different transportation modes. Comfort and pleasure indicator is analyzed through access to information, quality of the service, and security.

### 4.4. Analysis and Results

This subsection reports the results of the study to characterize the sustainable mobility initiatives. The study applies a urban data analysis approach, which has been also applied by our research group to study public transportation and other services in Montevideo [39,40]. The analysis accounts for relevant data about each initiative, obtained from open data sources (e.g., Open Data Catalog from the national government), data from previous studies (e.g., the urban mobility survey of Montevideo [38]), and also from personal interviews with both technicians of the local administration of Montevideo and managers of the companies that operate the studied initiatives (CUTCSA and Grin).

Quantitative Indicators

*Coverage*. The electric bus operated in several lines of CUTCSA company during 2017–2019. Table 1 summarizes the number of days of operation on the most relevant lines that operated the service. The percentage value for the number of days is also reported.

**Table 1.** Lines operated by the electric bus service in Montevideo (2017–2019).

| Line | Days | Percentage |
|------|------|------------|
| 128 | 78 | 14.0% |
| 142 | 16 | 2.9% |
| 169 | 47 | 8.4% |
| 180 | 303 | 54.4% |
| 181/183 (circular line) | 45 | 8.1% |
| 187 | 20 | 3.6% |
| other lines | less than 6 days | less than 1% |

According to the results in Table 1, the area considered to calculate the coverage of the electric bus service is the one corresponding to the buffer area defined by parallel segments located at 400 m of the most used lines routes: 128, 169, 180, and 181/183. The distance of 400 m is defined based on the recent mobility survey for Montevideo [38], which indicates that a person is willing to walk for up to about five minutes (corresponding to 400 m at a walking speed of 5 km/h) to access to a bus stop in order to use the public transportation service. In turn, for public bicycles, the coverage of the actual service and the projected coverage of the service are reported. The overall area for electric scooters is the one defined by the Grin service, which covers the area of service of the other two companies (Lime and Movo) that provided the service up to December 2019.

The area covered by each studied sustainable mobility initiative in Montevideo and the value of the *cov* indicator is reported in Table 2. Results were computed based on open data from each service.

**Table 2.** Coverage and the *cov* indicator for sustainable mobility initiatives.

| Initiative | Area | Coverage | Coverage Indicator |
|:---:|:---:|:---:|:---:|
| electric bus | 51.4 km$^2$ | 25.7% | 2.57 |
| public bicycle | 3.5 km$^2$ | 1.75% | 0.175 |
| public bicycle (projected) | 13 km$^2$ | 6.5% | 0.65 |
| electric scooter (Grin) | 23.5 km$^2$ | 11.75% | 1.175 |
| electric scooter (Lime) | 15 km$^2$ | 7.5% | 0.75 |
| electric scooter (Movo) | 7 km$^2$ | 3.5% | 0.35 |
| electric scooter (overall) | 23.5 km$^2$ | 11.75% | 1.175 |

The coverage maps for electric bus, public bicycles, and electric scooters services are presented in Figure 2. The analysis of the coverage indicator demonstrate that the area of service of each sustainable mobility initiatives is represents a small fraction of the total area of the city. The best coverage result was obtained for the electric bus service, which covers 25.7% of the city.

Coverage results are somehow expected, as the studied initiatives are new and public bicycles were introduced mainly for tourists. For electric scooters, coverage is also limited to zones with highest income (coastal area). Overall, the three studied modes provides a service that covers an area of 67.6 km$^2$, which represents 33.8% of the urbanized area of Montevideo, for a coverage index of 3.8. In conclusion, two-thirds of the the citizens who live in the urbanized area are not covered by these sustainable modes of transportation.

*Access to mobility service.* The population served by each service was computed by intersecting coverage areas with the population map and counting the total population in each zone. Figure 3 presents a superposition of the coverage area of sustainable mobility initiatives and the base map of population for Montevideo, grouped by census segments, which are used as the main administrative division for the Continuous Household Survey, from National Institute of Statistics, Uruguay [37]. The urban population of Montevideo is 1,305,082.

The electric bus service covers 429,269 citizens (32.9% of the population), accounting for the largest access index ($am = 3.29$). Public bicycles cover 86,917 citizens ($am = 0.67$) and the planned expansion is set to cover 193,368 citizens ($am = 1.48$). The electric scooters companies provides service to 285,445 citizens ($am = 2.19$). Overall, sustainable transportation modes cover 554,172 citizens (42.5% of the population, $am = 4.25$). As a consequence, the main conclusion from the analysis is that most of the urban population of Montevideo have no access to these sustainable modes of transportation.

*Affordability of sustainable mobility.* The affordability index was computed for the three studied transportation modes considering two types of trips: (i) short trips, with a length of 15 min, which is a reasonable traveling time for bicycles and and it is also the most frequent travel duration for scooters, according to the collected information, and (ii) long trips length of 45 min, which is the average time traveled in bus, according to the mobility survey [38].

Income per capita in Montevideo is USD 691 (middle income) and USD 423 (minimum income), as from data from August, 2019, and considering 1 USD = 37 Uruguayan pesos. On the one hand, electric bus applies a flat rate. The cost of a standard ticket, allowing one transfer trip in one hour, is 0.85 USD. On the other hand, electric scooters and public bicycles apply a time-based fare. The cost of using the public bicycles is 0 USD (free service) up to 30 min, and after that the rental cost is 0.74 USD for 30 min. For the electric scooter, the cost of a 15-min rent (the average time of utilization, as computed from the available data) is 2.1 USD and for one hour is 5.4 USD.

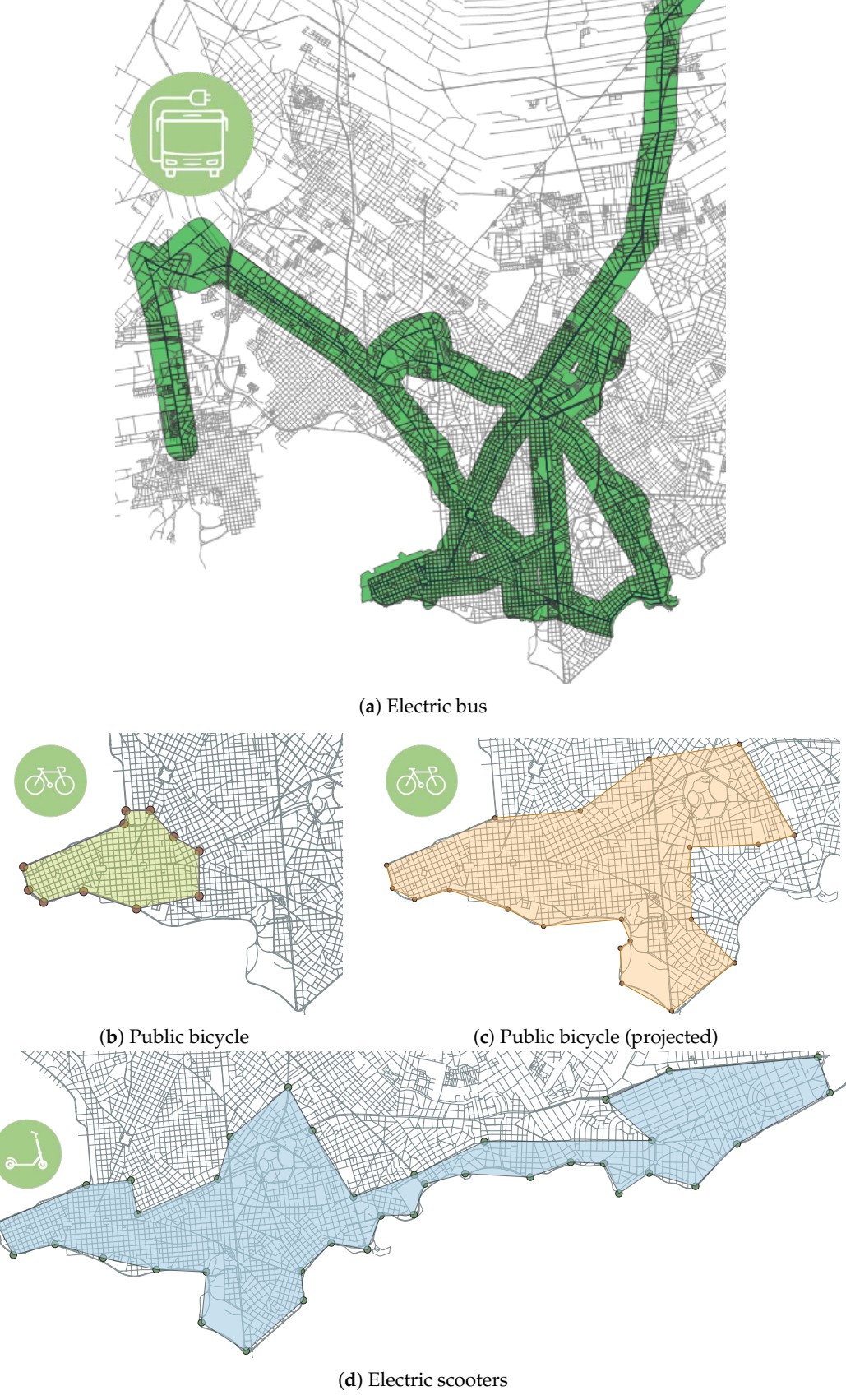

(**a**) Electric bus

(**b**) Public bicycle                    (**c**) Public bicycle (projected)

(**d**) Electric scooters

**Figure 2.** Coverage of sustainable mobility initiatives in Montevideo.

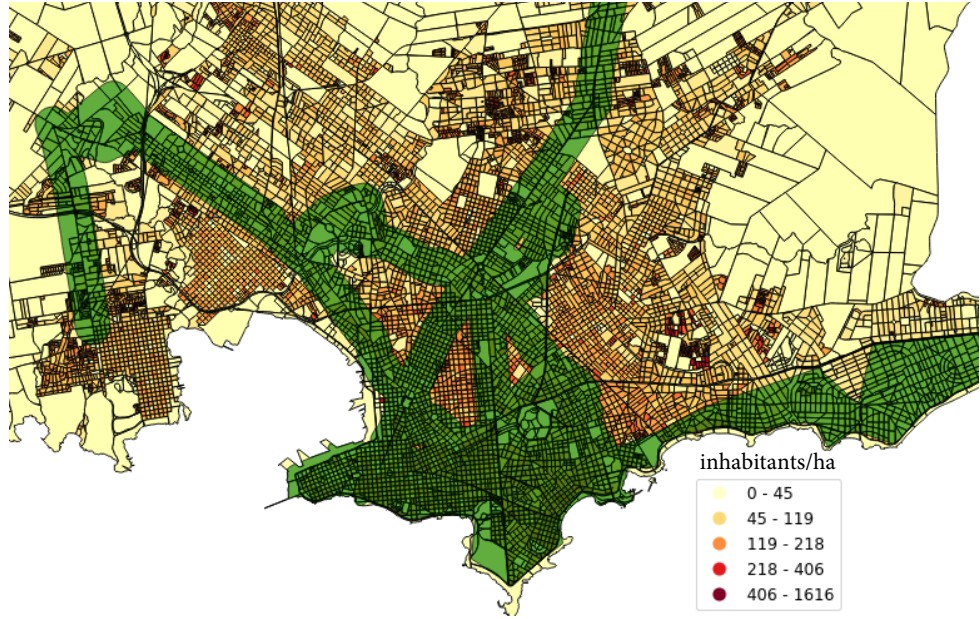

**Figure 3.** Coverage area of sustainable mobility initiatives and population of Montevideo (grouped by census segments).

Table 3 reports the affordability index of each sustainable transportation mode for middle and minimum income people.

**Table 3.** Affordability (*af*) indicator for minimum and middle income in Montevideo.

| | **Trip Length: 15 min** | | | | | |
|---|---|---|---|---|---|---|
| **income** | **45 trips** | | | **60 trips** | | |
| | bus | bicycle | scooter | bus | bicycle | scooter |
| minimum | 9.1% (8.2) | 0 (10.0) | 22.7% (3.9) | 12.0% (7.3) | 0 (10.0) | 30.2% (1.5) |
| middle | 5.5% (9.5) | 0 (10.0) | 13.9% (6.7) | 7.3% (8.8) | 0 (10.0) | 18.5% (5.2) |
| | **Trip Length: 45 min** | | | | | |
| **income** | **45 trips** | | | **60 trips** | | |
| | bus | bicycle | scooter | bus | bicycle | scooter |
| minimum | 9.1% (8.2) | 4% (9.8) | 57.2% (0.0) | 12.0% (7.3) | 5.4% (9.4) | 76.2% (0.0) |
| middle | 5.5% (9.5) | 2.5% (10.0) | 35.0% (0.0) | 7.3% (8.8) | 3.3% (10.0) | 46.7% (0.0) |

Results in Table 3 indicate that for 15 min trips, public bicycle has the maximum *af* value (10) for both income groups, as it is a free service up to 30 min. Affordability of bicycles does not reduce significantly when considering 45 min trips, due to the low fare of the service. Buses are cheaper than scooters for both short and long periods of time. Furthermore, the *af* indicator for buses is the same for both type of travels considered, while electric scooters downgrade to *af* = 0.0 for one hour trips. Overall, public bicycle is the most affordable transportation mode.

*Commuting travel time.* Table 4 reports the commuting travel times for three relevant distances for citizens' mobility in Montevideo: (i) 3 km, which is consider a short distance for those who primarily walk or ride a bicycle to work; (ii) 10 km, which is consider an average distance for bus commuters according to data from the urban mobility survey for Montevideo [38] and in line with similar studies for similar cities in the world [41,42]; and (iii) from end-to-end (*EtoE*) of the coverage areas for each mobility service. Speed and average times for bus were computed according to the methodology by Massobrio and Nesmachnow [40], using the Open Street Map service, estimations of average speed, and public applications available for the studied initiatives.

**Table 4.** Commuting travel times (min).

| Bus | | | Bicycle | | | Electric Scooter | | |
|---|---|---|---|---|---|---|---|---|
| 3 km | 10 km | EtoE (17.3 km) | 3 km | 10 km | EtoE (3.5 km) | 3 km | 10 km | EtoE (17.5 km) |
| 17.8 | 49.3 | 116.0 | 13.3 | 44.4 | 15.6 | 17.0 | 52.0 | 89.5 |

Results in Table 4 indicate that bicycle is the fastest option for both short (3 km) and long (10 km) distances, followed by the bus, and in third place the electric scooter. Differences between bicycle and bus reduce for trips of 10 km. EtoE bus trips takes longer than traveling on scooter, and almost the same time for shorter distances.

*4.5. Qualitative Indicators*

*Net public finance.* The electric bus initiative has received benefits from three subsidies in order to reduce the ticket price: a subsidy from the city administration to implement reduced fees for students and retirees, a fuel subsidy from the Ministry of Transportation, and other contributions from the Ministry of Economy and Finance. Furthermore, in 2019, bus transportation companies were granted a total of 100,000 USD each to promote the substitution of 4% of diesel buses to electric.

The public bicycles service is completely financed by the city administration of Montevideo to promote active and sustainable mobility. Finally, electric scooters do not received any subsidy as they are run by private companies.

*Energy efficiency.* All the studied transportation modes use clean renewable energy. Public bicycle is the most efficient of the initiatives, as it does not requires energy of external sources. Electric buses provides a significant improvement over diesel vehicles regarding energy efficiency. They produce no $CO_2$ emissions and have an iron phosphate battery that consumes 100 KWh each 100 km, which is a good rate for public transportation. Regarding electric scooters, the energy of operation represents a very low percentage of the total emissions generated (e.g., 4.7% according to the study by Hollingsworth for the city of Raleigh, North Carolina [43]). However, several other concerns arise, such as the non-clean energy required for collecting and distributing scooters, and the short life cycle of batteries, which can have negative environmental impacts. Even though the company introduced scooter stations to avoid picking up scooters one by one, users continue leaving scooters anywhere (the company did not apply any penalty fee for not using the stations).

*Intermodal connectivity.* The studied sustainable mobility initiatives operate in a common area of 2.8 km$^2$ (considering the projected expansion for the public bicycles system, the area increases to 7.3 km$^2$). Within this common area, public bicycles offer full connectivity with buses and scooters, as stations are located less than 100 m of bus stops and scooters are available nearby. Electric scooters facilitate door-to-door mobility, allowing users to leave scooters in specific stations or even anywhere within the operation area, thus providing a valid alternative for intermodal connectivity. Buses also allows intermodal connectivity, but it is limited to a few bus stops that have bicycles or scooters stations nearby.

*Intermodal integration.* Even though the three transportation modes studied provide intermodal connectivity, the system as a whole lacks of intermodal integration. Each service focuses on their own operation, without facilitating integration with others: no information or route guidance is provided to users, terminal bus stations do not provide parking lots for public bicycles or scooters, etc. The only effective integration is regarding the payment method for buses and public bicycles, which can be paid using the same public transportation card (STM). All these facts are specific drawbacks for intermodal mobility. Overall, integration should be improved to provide efficient mobility.

*Comfort and pleasure.* Available information of public buses (e.g., via mobile applications) is recognized as one of the best features offered to citizens, according to the recent mobility survey for Montevideo [38]. On the other hand, trip comfort (43.9%) and bus stop comfort (46.4%) are the worst rated attributes of the bus system.

Users have presented claims about the poor service of Movete and bad conditions of bicycles [44]. Furthermore, Montevideo lacks of a proper infrastructure (e.g., exclusive bicycle lanes) for connecting stations of the system. Although the city administration planned to expand the network of bicycle lanes, even in the expanded configuration they will be not enough to properly satisfy the needs of an increasing number of users. In addition, it is difficult to complete even small infrastructure modifications, such as the case of the bicycle line in Parque Rodó neighborhood, which is commented in Section 5.

Finally, users perceive many benefits of electric scooters: they are easy to locate, ride effortlessly, dock-less, and can be parked anywhere. On the other hand, electric scooters are vulnerable to road risks, as they are driven on the same lane as automobiles, and are an uncomfortable transportation mode for bad weather conditions.

*4.6. General Recommendations for Sustainable Mobility Initiatives in Montevideo*

This subsection provides some recommendations and suggestions that can be implemented in the city of Montevideo to promote sustainable mobility. Recommendations and suggestions are based on the review, analysis, and main results of the study of the three initiatives for public sustainable mobility, reported in the previous subsection.

One of the main facts observed from the analysis is that the initiatives for sustainable mobility are not widespread through the city. Instead, they provide a limited coverage and poor access to citizens. In this regard, one of the main recommendation is related to expand the coverage area, by introducing more bicycle stations, operating new lines of the electric bus, covering different routes or extending the routes offered, and expand the areas available to operate electric scooters. To improve coverage, more vehicles must be introduced and an articulated network of exclusive lanes has to be designed and implemented, which will help to improve other indicators too.

Specific suggestions to increase accessibility are extending the bus and bicycles network, and also the electric scooters operation. The expansion requires a proper previous evaluation of the real demand for each transportation mode, via direct methods (surveys) and indirect methods (mobility data analysis approach). Another suggestion to increase accessibility is to perform a viability study of offering the studied mobility services to medium and low-income areas, thus increasing the social impact of the initiatives. The proposed suggestions are in line with the strategies for sustainable mobility by bus reviewed by Fernández and Fernández [45], and also with the development of similar initiatives in Latin America.

Concerning affordability, the study demonstrated that electric bus is expensive and electric scooters are prohibitive for low-income citizens. This is a critical issue, mostly considering the periodic fare increases for those services at least once a year. In this regard, a specific suggestion for mobility services is to provide ticket packages for frequent users, and offering a lower price for combinations with other services, to facilitate inter-modality. Public finance support can be reviewed to better contribute to affordability, mainly by redirecting the assistance to reduce operation and maintenance costs, to guarantee a lower price for each service.

Several other suggestions are related to improve travel time, in order to provide more useful and efficient sustainable transportation systems. In this regard, both city administration and transportation companies must focus on providing accurate information to citizens and guaranteeing a quick access to relevant information for travel planning. Electric bus should provide a higher frequency service, by redesigning or updating existing timetables, and a better effort must be done in order to provide good synchronization between different bus lines. For public bicycles and electric scooters, travel times are related to the availability of vehicles and also on the available interconnection network, so specific improvements on the fleets size and on infrastructure can contribute in this regard.

To take advantage of the modal shift from diesel to electric buses to improve energy efficiency, smart planning of battery charge is needed, by properly locating charge stations in strategic points of the operation area or planning the use of external batteries. Electric scooters also need to review their

operation efforts for collecting and distributing vehicles, which currently demands non-clean energy. A specific suggestion to improve efficiency is installing secure parking stations to charge scooters batteries while parked.

A clear recommendation to enhance sustainable mobility is to promote intermodal connectivity between transportation modes. In this regard, services should work on providing real-time data information (e.g., vehicles available, location, bus stops information, timetabling, etc.) and on installing shared stations for at least two of the studied services. A specific suggestion is to integrate the ticketing system, allowing users to share modes within a ride, maybe linked with the aforementioned offers to improve affordability.

In terms of comfort and pleasure, companies can offer a better quality service by improving the comfort of the vehicles, and particularly adopting security measures to guarantee safe travels. Bicycles and electric scooters can incorporate helmets to their service and buses can include seat belts for passengers. Related to the overall quality of experience, companies and city administration can improve access to information providing users with mobile applications oriented to reduce walking time, waiting time, and the overall travel times.

Other mobility suggestions regarding relevant features such as age, gender, socioeconomic situation, etc. can be performed when proper data is available, in order to extend the overall analysis in the mobility survey [38]. We are working to get that information from the city administration (Intendencia de Montevideo) under current project "Spatial, universal, and sustainable accessibility: characterizing the multimodal transport system of Montevideo, Uruguay".

In general, the economic viability of the proposed suggestions is feasible within the current business models of the companies that operate each service. Furthermore, most of the suggestions are in line with current developments by national institutions (city administration, Ministry of Industry and Energy), which have committed funds for promoting and developing sustainable transportation and sustainable mobility in the city.

## 5. Practical Approach for Analysis and Implementation of a Sustainable Mobility Plan for Engineering Faculty and Parque Rodó Neighborhood, Montevideo

This section presents a specific case study that demonstrate the viability of analyzing and implementing a sustainable mobility plan for Engineering Faculty and Parque Rodó neighborhood, Montevideo.

### 5.1. Mobility Analysis and Survey

This subsection describes the studied area and the methodology for collecting and analyzing mobility data.

#### 5.1.1. Engineering Faculty and Parque Rodó Neighborhood

Engineering Faculty (Facultad de Ingeniería) is the school in charge of engineering and other technology-related studies within Universidad de la República, Uruguay. In 2020, the Engineering Faculty has 10,350 students, 915 professors, and 195 administrative employees [46]. All these persons have specific mobility demands to access to the institution.

Engineering Faculty is located in Parque Rodó neighborhood (South of Montevideo). A map of the studied area is presented in Figure 4. The studied area covers $0.5\,\text{km}^2$ and includes three main avenues: Herrera y Reissig, where Engineering Faculty is located; Sarmiento; and Sosa. Nearby the Engineering Faculty is Aulario Massera, a large classroom building shared by Architecture, Economics, and Engineering faculties.

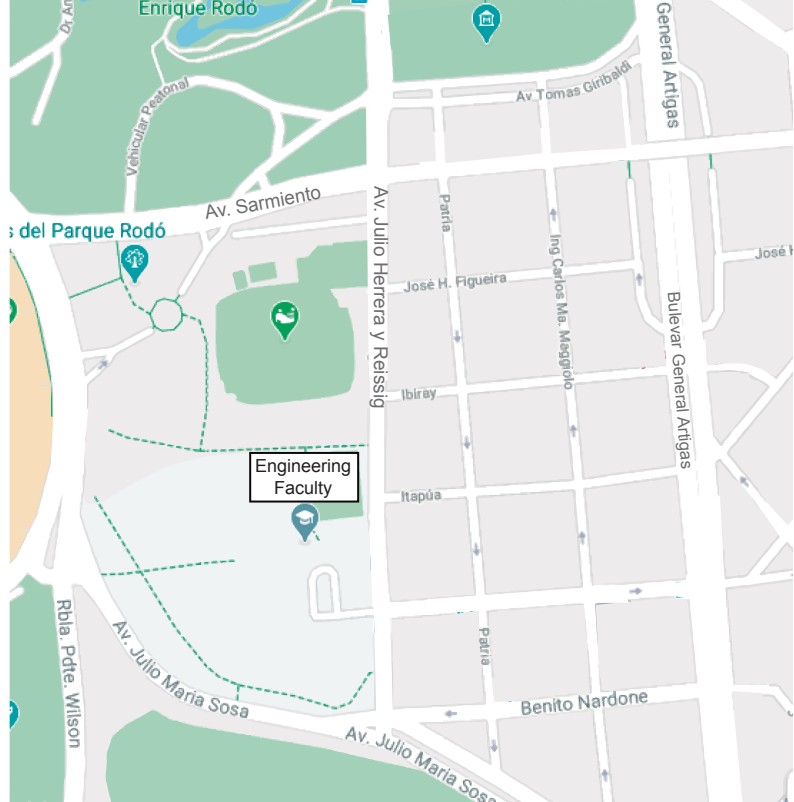

**Figure 4.** Area considered in the study: Engineering Faculty and Parque Rodó neighborhood.

Engineering Faculty has two parking lots with parking capacity for about 140 vehicles. The building also has bicycle parking (open from 7:00 to 23:00 from Monday to Saturdays) with security monitoring and a parking capacity of 330 bicycles. The bicycle parking has restrooms with showers and lockers to promote students using their own bicycles for traveling. This facility is under current norms for bicycles parking in public institutions, according to the administration of Montevideo.

Engineering Faculty has been promoting sustainable mobility initiatives. On June 2004, a group of professors founded "Unibici", a program to promote the use of bicycles between students. Moreover, Engineering faculty worked together with the city administration of Montevideo to create bike lanes in a circuit connecting faculties of Universidad de la República. However, the project has not been completed yet.

### 5.1.2. Motivation and Objectives of the Study

The main motivation of the study is to understand the mobility demands to Engineering Faculty and Parque Rodó neighborhood, and also from Engineering Faculty surroundings to other zones of the city. This is a relevant case study, which includes a variety of interesting features: Parque Rodó is a residential area, but also has a high education center (and others in the surrounding area), a shopping center nearby, several health centers in the zone, and other services. The objective of the study is to identify, analyze, and characterize the current situation regarding mobility and sustainable mobility in the studied zone, for different groups of people. This is a different study to the one performed to characterize sustainable transportation and sustainable mobility initiatives in Montevideo (reported in Section 4). In the case study in Parque Rodó, the study is based on data collected in situ and the opinions of interviewed people are taken in consideration.

The analysis of the current mobility situation provides quantitative and qualitative information for a systematic characterization of mobility demands in Engineering Faculty and Parque Rodó neighborhood. Moreover, the survey allows determining if the studied groups of people would be

willing to change to more sustainable transportation modes and the specific issues that prevent them to make that change.

5.1.3. Methodology for Collecting Data

The methodology applied for gathering mobility information on the studied area consisted in collecting the information of the universe of study and performing a survey in situ.

Four relevant groups of people were identified: (i) students of Engineering Faculty and other faculties that shares Massera classroom building, (ii) professors and employees of Engineering Faculty, (iii) people who live in the neighborhood, and (iv) people who work on the neighborhood.

The total number of people involved in the analysis was 617 (79 living in the area + 538 commuting from other zones of the city). Thus, the study considered a sample size of 2.15% for the analysis of the mobility situation of Parque Rodó neighborhood. The estimated size of the relevant universe is 28,602 persons, including people that live in the studied zone; students, professors, and employees of Engineering Faculty; and persons that commute to the area from other zones of the city. The sample size considered in the survey is significantly larger than the one used is similar initiatives. For example, the mobility survey for Montevideo [38] studied 2230 homes, interviewing a total number of 5946 persons, which represent a sample size of 0.4% of the urban population of Montevideo.

By considering not only Engineering Faculty, but also the surrounding neighborhood, the survey intends to capture a more holistic view, taking into consideration the different groups of people that travels to/from the studied area.

A survey was formulated to know the mobility characteristics of the studied groups of people. The survey included the following questions.

1. Do you study or work at Engineering Faculty?
2. Do you travel often to this area?
3. What is the origin and destination of your trip?
4. What transportation mode(s) do you use for commuting to study/work in the neighborhood or from this neighborhood to other zones of Montevideo?
5. If you use more than one transportation mode, specify the percentage of utilization.
6. How often do you make these travels weekly?
7. Which aspects are the most relevant for you while commuting?
8. Would you be willing to switch to a more sustainable transportation mode?
9. To what transportation mode would you be willing to change?
10. What do you think it prevents you to change to a more sustainable mobility ?

The survey was performed face-to-face to people circulating in the studied area. Interviews were performed in different locations, including the front door of Engineering Faculty, five bus stops located less than 300 m of the faculty, a bakery located 100 m from the faculty, the front door of Franzini football stadium, and also in random locations at streets in the zone: Julio Herrera y Reissig, Itapua, Ibiray, Patria, José Figueira, Eduardo Garcia de Zuñiga, Benito Nardone, Julio Maria Sosa, Carlos María Maggiolo, Sarmiento, Senda Nelson Landoni, and Bulevar Artigas. People were not interviewed at home, because the main interest was in specific mobility demands (e.g., people attending to Engineering Faculty, moving from/to work, or moving to shops in the area).

The questionnaires were performed during 15 November–15 December 2019, from Monday to Friday, from 8:00 a.m. to 7:00 p.m. Weekend trips were not considered in the analysis because they are significantly lower than working days trips. Engineering Faculty offers just a few classes on weekends (just on Saturday morning, for some sporadic activities) and commercial activity in the studied zone is also reduced on weekends. People who commute in sustainable transportation modes were not asked if they would be willing to change towards a more sustainable transportation, as they already do it. The study also gathered information of bus lines that operates in the zone and identified the bus stops near the faculty. Scooter stations and bicycle lanes were also identified.

5.1.4. Methodology for Data Analysis

The study applies a urban data analysis approach, accounting for relevant data from the survey and also information from public sources.

Regarding the methodology applied for data analysis, the study analyzes global characteristics of mobility demand in the area. Some indicators used for the global case of Montevideo are studied, e.g., coverage and commuting travel time, as defined in Section 4.3. In addition, other relevant aspects related to the sustainable mobility characterization are analyzed, such as travel distance and modal-choice preferences for trips. Travel distance is defined as the distance that a person travels from any point of the city to the centroid of the Parque Rodó neighborhood. All distances are computed using the Google Maps service. Modal-choice preference of commuters is defined as the decisions taken by individuals to chose one transportation mode instead of another. The reason for the choice is linked to several factors, including affordability, travel time, comfort, accessibility, and sustainability.

Furthermore, the study analyzes the quality of service of existing mobility options through mobility preferences while commuting, such as cost, comfort, speed, security, sustainability, and other valuable interests for citizens. The studied mobility preferences may not correspond to the transportation mode that people use today, but to modes that they are willing to use if those preferences and related issues improve.

Some indicators analyzed in the case study of Montevideo are not taken into account in the study of Parque Rodó and Engineering Faculty. For example, affordability or access to mobility service indicators are not computed, mainly because of two reasons: (i) from the point of view of the price of mobility services, prices are the same for all zones in Montevideo, thus the main results reported in Section 4 also holds for Parque Rodó and Engineering Faculty neighborhood, and (ii) most of the studied universe consists of middle/high income people, which normally can afford all transportation modes (this fact is confirmed by the low number of trips from/to those zones of the city with the lowest income per capita, which is below 8%).

*5.2. Analysis of Results*

This subsection reports and discusses the most relevant results of the study. The most relevant results of the survey are presented on graphics, tables, and maps that allows characterizing distances, transportation modes, and other relevant features related to sustainable mobility in the studied zone.

*Coverage.* The studied area is fully covered by all the studied transportation modes (bus, bicycles, and electric scooters). Seven bus lines operates in the neighborhood, directly connecting people with many zones in the city. Furthermore, all locations in the city can be accessed via transfer trips. Although bus-only lanes were defined in main road and avenues of Montevideo, they are not defined in the studied area, so buses share the road with private transportation.

However, just 47 trips of the electric bus (3.6% of the total trips performed in 2016–2019) operated in lines that serve Parque Rodó neighborhood. Regarding bicycles, the current public system does not cover the studied area, but it is projected to be covered in the expansion, as reported in Section 4. Engineering Faculty provides the bicycle parking and other services for students, professors, and workers that use this transportation mode. Scooters operates all through the zone, having five stops near Engineering Faculty. Figure 5 present a coverage map of the studied zone, highlighting bus line routes and stops, scooter stops, bicycle parking areas, and bicycle lanes. The bicycle parking of Engineering Faculty is distinguished as it provides covered parking, security, and showers.

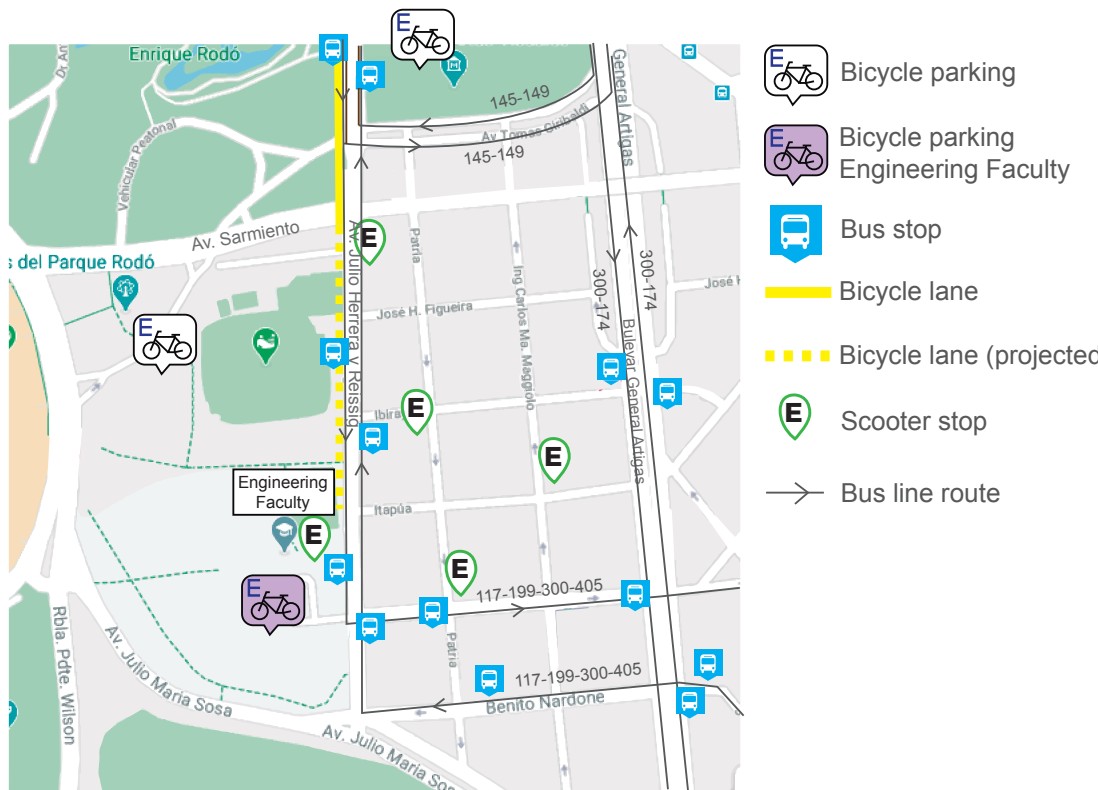

**Figure 5.** Coverage of Engineering Faculty and Parque Rodó neighborhood by the studied transportation modes.

*Transportation modes*. Regarding the transportation modes used by people commuting to/from Parque Rodó and Engineering Faculty, Table 5 reports the number of trips using each transportation mode declared in the survey and the percentage that it represents over the total. Transportation modes are listed from more sustainable to less sustainable.

**Table 5.** Transportation modes used for commuting to/from Parque Rodó neighborhood and Engineering Faculty.

| Transportation Mode | Number of Trips | Percentage |
|---|---|---|
| walking | 83 | 13.0% |
| bicycle | 40 | 6.3% |
| scooter | 0 | 0.0% |
| bus | 361 | 56.4% |
| more than one transportation mode (on different days) | 69 | 10.8% |
| non-sustainable transportation modes (car, motorcycle) | 87 | 13.6% |
| total | 640 | 100.0% |

According to the results reported in Table 5, just 19.3% of the trips to/from Engineering Faculty and Parque Rodó are done using sustainable transportation modes. Overall, more than half of the trips are done using the bus. The number of trips using other non-sustainable transportation mode is 13% (mainly private cars, just 1.4% on motorcycle), almost the same than people walking to/from the studied area. Bus is the most popular transportation mode, mainly because it is the most accessible and affordable transportation mode for large distances, as confirmed by the accessibility and affordability analysis of transportation modes for the city on Montevideo, reported in Section 4.4.

*Travel distances*. A summary of distances traveled by people from/to Parque Rodó and Engineering Faculty is reported in the pie chart in Figure 6. Travel distances were calculated in Google Maps considering the origin the Engineering Faculty and the destination the neighborhood people reported.

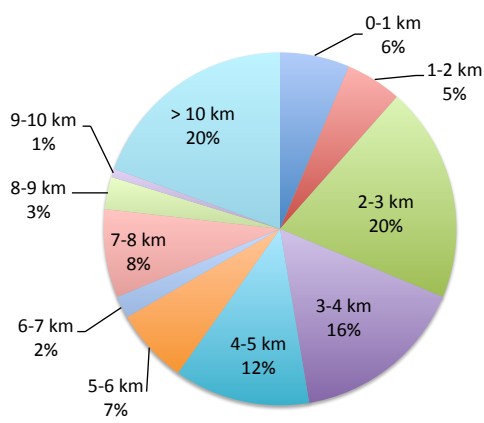

**Figure 6.** Travel distances according to data from the survey.

The analysis of travel distances indicates that 60% of the surveyed people commute from a maximum distance of 5 km away, and one-third of them travel between 2 to 3 km away. In addition, just 20% of the surveyed people commute a distance greater than 10 km. Furthermore, 95% of them declared to do a round trip, and 90% commute to the same place with a frequency of three times a week or more. These results confirms that the mobility demands in the studied zone follows a regular pattern, and that sporadic trips do not contribute significantly. Thus, the proposed approach, based on the analysis on frequent trips, provides a realistic characterization of mobility demands to/from Parque Rodó and Engineering Faculty.

*Transportation modes by distance.* Figure 7 refines the analysis of transportation modes, considering the average distance for each surveyed trip.

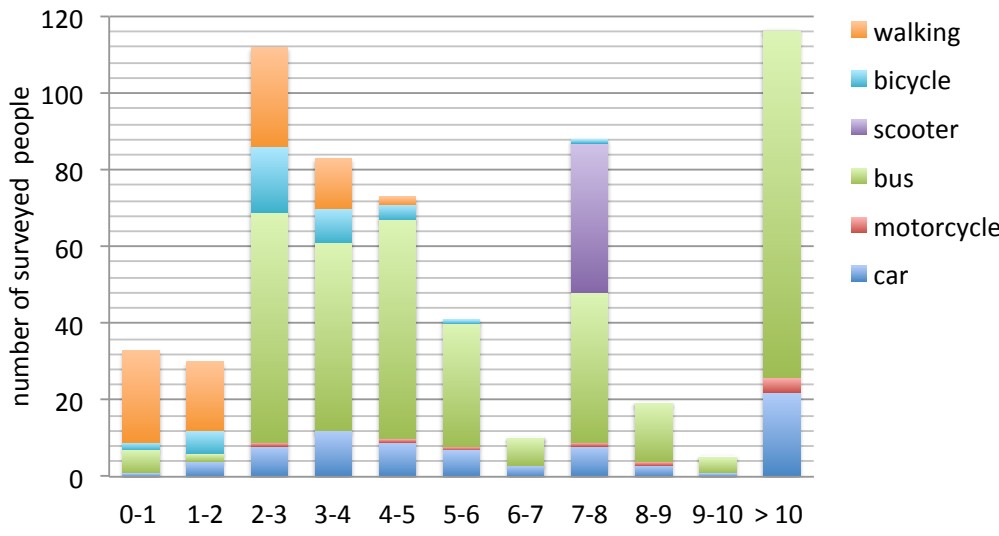

**Figure 7.** Transportation modes by distance.

The analysis of data reported in Figure 7 allows concluding that walking is the most popular transportation mode for distances less than 2 km, followed by bicycle and bus. For distances between 2 and 5 km, bus is the most popular transportation mode, followed by walking and bicycle. For distances longer than 5 km, bus is still the most used transportation mode, followed by non-sustainable transportation modes: car and motorcycle. Overall, the large number of people commuting to Parque Rodó and Engineering Faculty using non-sustainable transportation modes suggests that there is room to improve towards sustainable mobility in the studied area. Especially, 56.4% of trips using bus indicate that significant improvements to the service are definitely possible,

by using electric buses. Specific actions can be also proposed to consider people traveling on car and motorcycles. This issue is studied in the following paragraphs, considering the information about preferences and motivations collected in the survey.

*Commuting travel time*. The combined analysis of distance and transportation modes allows computing the average commuting travel times for people commuting from/to the studied zone. In this regard, Table 6 reports the average travel times from/to the five most demanded origin/destination of surveyed trips, grouped by neighborhoods of Montevideo. The distance for each neighborhood is measured from Engineering Faculty to the centroid of each neighborhood.

**Table 6.** Commuting travel time to Engineering Faculty/Parque Rodó from the most frequent neighborhoods as origin/destination of trips.

| Neighborhood | Distance | Bus | Bicycle | Scooter | Walking |
|---|---|---|---|---|---|
| Parque Rodó | 1.0 km | - | 4.4 min | 7.0 min | 12.0 min |
| Cordón | 2.5 km | 18.9 min | 11.0 min | 14.5 min | 30.0 min |
| Tres Cruces | 3.0 km | 21.2 min | 13.3 min | 17.0 min | 36.0 min |
| Pocitos | 3.5 km | 28.4 min | 15.5 min | 19.5 min | 42.0 min |
| Centro | 3.7 km | 24.4 min | 16.4 min | 20.5 min | 44.4 min |
| Prado | 8.0 km | 44.4 min | 35.5 min | 42.0 min | - |

Results reported in Table 6 indicate that bicycle is the fastest transportation mode from distances shorter than 3 km. Scooter and bus are second and third regarding travel times, respectively. Considering that bicycles have no cost (either for using private vehicles or the public service projected for the zone) for up to 30 min, the bicycle is the fastest, most affordable, and most sustainable transportation mode for short distances. For distances between 3 km and 8 km, bicycle is the fastest transportation mode too, but a relevant issue that must be taken into account: when traveling long distances, commuters must consider that they might need to shower/change clothes due to the physical effort required, which would demand from a few to 10–15 additional minutes. Then, bus takes approximately the same time (i.e., 10 min more, but with no need to shower/change clothes) and scooter requires about four minutes more than bicycle, but considering that users can leave scooters in any place and there is no need to shower/change clothes, it is the fastest transportation mode. For distances larger than 8 km, all the studied transportation modes takes approximately the same time, so it is reasonable that most people use the bus, which is the most comfortable transportation mode, as reported in the previous analysis of transportation modes by distance.

*Aspects people prioritize while commuting*. Figure 8 summarizes the results of the analysis of those aspects identified as more relevant for people while commuting. The reported results are not necessarily linked to the transportation mode people use today, but to aspects they prioritize when commuting.

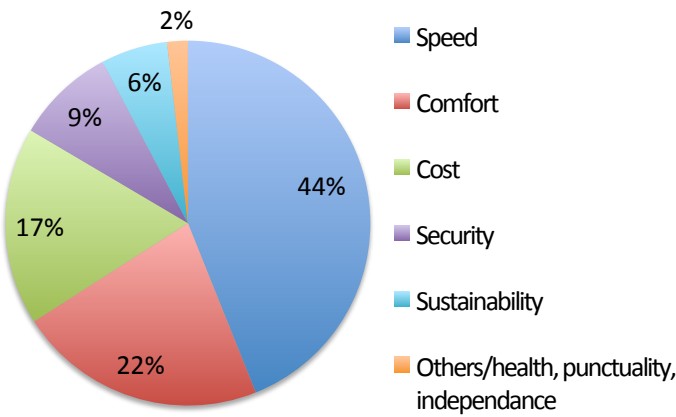

**Figure 8.** Aspects people prioritize while commuting.

The analysis of the aspects people prioritize while commuting indicates that 44% of the interviewed people prefer arriving faster to their destinations than other aspects. Comfort is the second feature more valued by the surveyed people (22%), and cost in third place (17%). Results obtained in the survey confirmed that aspects people prioritize do not depend on the distance or the travel time. In general, speed, comfort, and cost (in that order) are mentioned as priorities in declarations by surveyed people.

Overall, one conclusion can be formulated from the obtained results: public bus is the only transportation mode that could offer the three aspects people prioritize, in case it improves the actual service conditions. The other studied transportation modes cannot offer those three aspects, mainly because some of the are expensive (like private vehicles), ans several others require a physical effort and/or they are not comfortable in adverse climate conditions (like bicycle and scooter).

*Willingness to change towards more sustainable transportation modes.* The study interviewed 617 people, 504 of whom commute in non-sustainable transportation modes and 113 in sustainable transportation modes (bicycle or walking). Of those 504 people, 468 would be willing to change to sustainable transportation modes. This is a very relevant result, and is accounted as an empirical metric to determine the public acceptance of sustainable transportation modes within the people interviewed in our research. Furthermore, it is a first hint of the positive views towards sustainability, which can be confirmed by performing similar interviews in other (representative) neighborhoods of Montevideo. Figure 9 reports the results of the analysis of the sustainable transportation modes people would be willing to change.

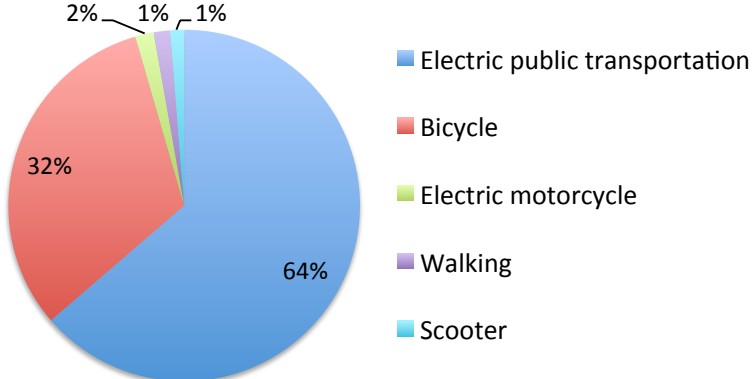

**Figure 9.** Sustainable transportation mode people who commute in non-sustainable transportation mode might change.

According to the results reported in Figure 9, electric public transportation is the mode that most of the people would be willing to change (64%), followed by bicycle (32%). Considering that CUTCSA and other bus companies plan to develop the modal shift from diesel to electric buses after evaluating the results of the pilot plan explained in Section 4, the willingness to change can provide a big leap in sustainable mobility in the studied area.

Taking into account the aforementioned results, the study analyzes next the reasons why people would like to change and why they do not actually change to both preferred transportation modes (electric bus and bicycle).

The reasons why people are willing to change to electric bus are mainly related to be part of initiatives oriented towards decarbonizing public transportation to reduce climate change and mitigate the environmental impacts of fossil fuels. Energy efficiency is also a motivation, especially considering that Uruguay is one of the leader countries in renewable energy in the world and it has a surplus of generated energy (over 98% of it generated from clean resources, according to reports for 2019 [47]). These opinions are in line with recognized benefits that electric public transportation provides to to the

communities they serve (improving air quality, reducing greenhouse gas emissions, financial benefits related to reduced maintenance and operating costs, and avoiding healthcare expenses) [48].

On the other hand, Figure 10 summarizes some reasons why people might not change to electric public transportation. The analysis considers that no modifications on the current routes and frequencies will be associated to the electric bus, which will operate on the same conditions of the actual service (as suggested by the pilot plan implemented by CUTCSA).

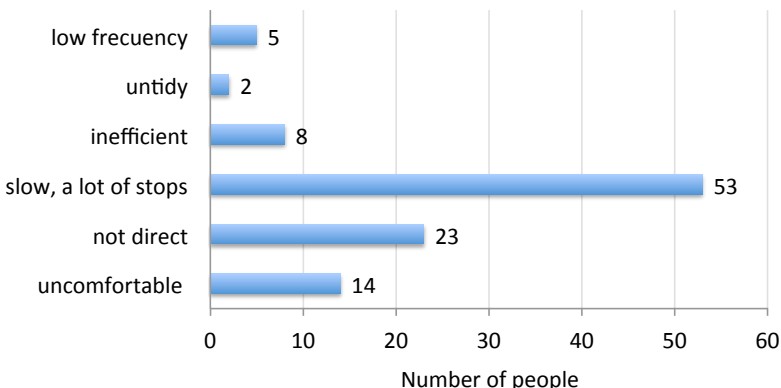

**Figure 10.** Reasons why people might not change to an electric public transport.

Regarding the results reported in Figure 10, the study collected opinions of 49 persons that travel in cars and would be willing to change to electric public transportation. However, most of them declared they will not change in case the bus will be inefficient, slow, untidy, with low frequency, and not direct. Additionally, 151 persons that travel today by bus would be willing to change to electric public transportation, even though most of them declared that they would also like to be faster, more comfortable, and more direct.

According to the survey, 149 persons reported that they would be willing to change to bicycle as transportation mode. In turn, 114 persons declared the reason why actually they do not use bicycle for commuting. The reasons why people are willing to do that modal change are related to the main benefits of riding a bicycle regarding health and also because it is the cheaper transportation mode, just as reported for the case of study of Montevideo.

On the other hand, Figure 11 summarizes the main reasons why people do not change their actual transportation mode to bicycle. Results correspond to 114 persons (eight traveling by car and 106 traveling by bus), who gave additional information about the reasons that prevent them to switch to bicycle. Car travelers mostly indicated the main reason is the reduced traffic safety, mainly because the few bike lines available in the city. In turn, bus travelers declared the main reasons for not using bicycle are the lack of proper facilities (e.g., their workplace does not offer a parking bicycle), they do not have enough space at home, and because they cannot afford a bicycle. Most actual bus travelers mentioned that they would use a public bicycle system if it was operating in the area.

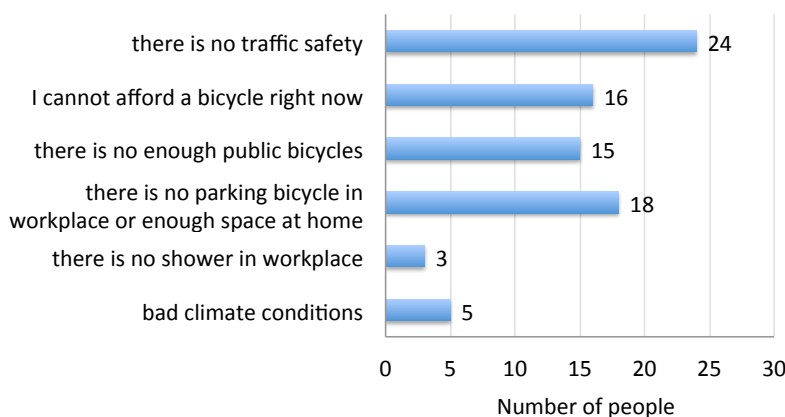

**Figure 11.** Reasons why people do not change to bicycle.

### 5.3. General Recommendations for Sustainable Mobility Initiatives in Parque Rodó Neighborhood and Engineering Faculty

This subsection provides specific suggestions and recommendations for improving sustainable mobility in the Engineering Faculty and Parque Rodó neighborhood. Recommendations and suggestions are based on the review, main results, and analysis of the mobility demands of the studied area, reported in previous subsections, especially considering the following concepts; the aspects related to accessibility (explored from the point of view of the infrastructures and services and and also from the point of view of people that commute from/to the studied area); the detected mobility patterns; and the motivation and opinions of interviewed people, which in fact constitutes a direct contribution of our research, as no previous similar studies have been developed in Montevideo.

One of the main facts observed from the analysis is that the studied area is fully covered by all the studied transportation modes (bus, bicycles, and electric scooters). This fact makes it easy connecting people with other zones of the city directly or via transfer trips. However, in terms of sustainable mobility, the neighborhood is not covered by the public bicycle initiative, which was analyzed for the case study of Montevideo, and the electric public transportation of the pilot plan of CUTCSA developed during 2016–2019 only performed 3.6% of the total trips traveled through the studied area. In this regard, one of the main recommendations is related to expanding the coverage area of public bicycles, by introducing bicycle stations in this area and design an articulated network of exclusive lanes, which also will help to improve other indicators, besides coverage.

Regarding the used transportation modes, more than half of the trips from/to the studied area are made by bus. For distances longer than 2 km, bus is the most popular transportation mode. Furthermore, 64% of the interviewed people that travel using non-sustainable transportation modes would be willing to change to electric public transportation. In addition, results of the study confirmed that people will not change to electric public transportation if the conditions of the service remain as nowadays. This is an important result because the pilot plan implemented by CUTCSA was developed in identical conditions than the actual service, regarding routes, bus stops, travel times, and other relevant indicators. In this regard, several suggestions are related to improve public transportation, in order to provide more useful and efficient sustainable transportation systems.

According to the commuting travel time indicator, some people declared they are not willing to change their actual transportation mode for electric public transportation because buses are very slow, they have many stops, low frequency, and routes are not direct. Thus, some suggestions for the new electric public transportation relate to introducing lines with fewer stops and higher frequencies than the current service to allow commuters, especially those whose trips demand more than 50 min (10 km), arrive faster to destination.

In terms of comfort and pleasure, bus companies can offer a better quality of service by improving the comfort of vehicles. Some ideas to give users a better service in terms of comfort

and pleasure include improving travel conditions (e.g., appropriate space, air-conditioning, and free WiFi), guaranteeing universal accessibility, providing accurate real-time information via mobile applications and digital screens in bus stops, reducing motor vibration and noise, among others.

Regarding infrastructure, in 2010 Montevideo incorporated bus-only lanes in main roads and avenues, to avoid traffic congestion and speed up public transportation. However, bus lines that circulate through the studied area still share the same lane with other transportation modes. In this regard, the studied area has few wide avenues to install bus-only lanes; e.g., Herrera y Reissig, which crosses Parque Rodó neighborhood, has only one line in each direction from Sarmiento to Sosa (end of the avenue), thus including a second (bus-only) line would require a major infrastructure modification. However, mobility can still benefit for installing bus-only lines in Bulevar Artigas or in Herrera y Reissig (north, where there is space available). Regarding infrastructure for bicycle, considering the surveyed responses, two relevant suggestions are formulated to foster the modal shift: (i) bicycle lines should be extended, at least to include the projected line that will reach Engineering Faculty (which is planned since 2013, and has not been constructed due to non-disclaimed reasons); (ii) in addition, companies located in the zone and also in the main destination neighborhoods should be encouraged to provide bicycles parking within workplaces and also restrooms with showers, to be used by employees after the physical effort required for a ride.

The reported results, descriptive statistics, and suggestions are very valuable for the city administration in order to conceive an effective sustainable mobility plan in the studied area.

Finally, we acknowledge the implicancies of the reported analysis on policies and decision-making related to two relevant research and development initiatives our research group is currently participating on: (i) local sustainable mobility plans, developed by Ministry of Industry, Energy, and Mining through the MOVES project, and (ii) project "Spatial, universal, and sustainable accessibility: characterizing the multimodal transport system of Montevideo, Uruguay", developed with the support of the local administration (Intendencia de Montevideo), with the main goal of creating valuable knowledge and formulate specific policies to develop and improve mobility and accessibility. Some specific examples that can benefit from the analysis reported in the previous subsection for Parque Rodo neighborhood are the redesign of bicycle lines in the zone and the planning of a route for the electric bus to provide mobility services to Engineering Faculty and other faculties in the district (Architecture Faculty and Economics Faculty).

## 6. Conclusions and Future Work

This article studied sustainable mobility initiatives implemented in Montevideo, Uruguay, and a specific case study following a practical methodology to characterize and improve sustainable mobility in the Parque Rodó neighborhood and Engineering Faculty.

The study analyzed the main concepts of sustainable mobility by a review of related work on the topic and applied the existing knowledge to analyze three sustainable transportation modes currently available in Montevideo (electric bus, public bicycles, and electric scooters) through quantitative and qualitative indicators of sustainable mobility. Results of the study confirmed that the coverage area of the studied sustainable mobility initiatives is a small fraction of the total area of the city, thus a significant part of the population of Montevideo cannot access to sustainable transportation modes. Regarding cost, public bicycle is the most affordable mode of transportation, and electric bus is the second best option, mainly because these two services benefit from subsidies and support from public finances, thus they can keep a reasonable price for users. Electric scooters have prohibitive prices for low-income citizens. Public bicycle is also the fastest and the most ecological option for short and long distance travels. On the other hand, the quality of service of the public bicycle system, regarding comfort and pleasure, is the worst of the three studied transportation modes. Public bicycle, electric bus, and electric scooter provide intermodal connectivity between them, but there is a lack of intermodal integration between services. Specific suggestions were provided in regard of the main drawbacks of current sustainable mobility initiatives in Montevideo.

The mobility analysis of Parque Rodó neighborhood and Engineering Faculty was based on a survey performed to 617 persons who commutes to/from the neighborhood from/to other zones of the city. This is an important contribution of the reported research, since no previous analysis of sustainable mobility has been performed for specific zones of Montevideo.

Results of the study indicate that the area is fully covered by all the studied transportation modes (bus, bicycle, and electric scooters). However, in terms of sustainable mobility, the neighborhood lacks of a proper coverage, as the public bicycle initiative does not operate in the area and electric public transportation only did 3.6% of the total trips in 2016–2019 through the studied area. The survey reported that more than half of the trips from/to the studied area are made by bus. Bus is also the most popular transportation mode for distances longer than 2 km. This is a relevant result, because, according to the survey, more than half of the persons that currently travels using non-sustainable transportation modes would be willing to change to electric bus. Thus, improving the bus public transportation service towards the modal shift to electric emerges as a first priority.

On the other hand, despite the fact that the pilot plan for electric buses was successfully deployed and valued by citizens, and this transportation mode will provide users a better service and move towards sustainable mobility, people declared they are not willing to change their actual transportation mode if electric buses operate in the same conditions to the actual service. As a consequence, the shift from diesel to electric does not ensure the modal change to public sustainable transportation if buses still operate as in the actual service. To attract more users, bus companies should work on improving speed (one of the most important aspect people prioritize while commuting) by rethinking the routes and stops of new electric buses and also travel conditions in terms of comfort and pleasure (e.g., providing appropriate space, air-conditioning, free WiFi, etc.), which is another of the main aspects to solve to succeed in the modal change of citizens.

An important result of the mobility analysis in Parque Rodó and Engineering Faculty is that more than half of the surveyed trips involve distances shorter than 5 km, i.e., suitable distances to commute by bicycle. The study also reported that a quarter of the interviewed people are willing to switch to bicycle as a sustainable mobility alternative. However, the lack of infrastructure (bicycle lanes, bicycle parking and showers at workplace) discourages people to use the bicycle for commuting to/from the studied zone. Thus, improving the mobility conditions for bicycles turn to be an important topic to address in the studied zone.

Overall, the analysis of the three sustainable initiatives in Montevideo and the study of the sustainable mobility situation and demands of Parque Rodó neighborhood and Engineering Faculty are valuable tools for helping academics, transportation companies, and stakeholders to analyze and evaluate possible solutions to implement sustainable mobility plans. The reported case study in Parque Rodó provides a basis for building more powerful surveys and data collection activities to better understand sustainable mobility in the whole city.

The main lines for future work are related to extend the analysis, by including some studies that were not taken in consideration, e.g., discussion of automobility in the city (traffic volume data, number of traffic accidents, level of air pollution) and the measures that the city of Montevideo has taken to overcome the problems generated by the automobile, building specific origin–destination maps to account for the most demanded travels to/from Parque Rodó neighborhood and performing a deep study of the reasons why people would be willing to change to sustainable transportation modes and a deeper analysis of public acceptance via global interviews. The analysis of differential mobility needs is also an important line for future research, to assess the impact of sustainable mobility in Montevideo, focusing on its citizens. Regarding the replicability of the study, the proposed methodology can be applied to characterize the mobility demands and the sustainable mobility analysis on other relevant neighborhoods in the city. Furthermore, the analysis can be also extended by considering other sustainable mobility indicators and studying best practices implemented on other cities, in order to contribute to the improvement of urban sustainable mobility in Montevideo.

**Author Contributions:** Conceptualization, S.H. and S.N.; methodology, S.H. and S.N.; software, S.H. and S.N.; validation, S.H. and S.N.; formal analysis, S.N.; investigation, S.H. and S.N.; resources, S.H. and S.N.; data curation, S.H.; writing–original draft preparation, S.H. and S.N.; writing–review and editing, S.N.; visualization, S.H. and S.N.; supervision, S.N.; project administration, S.N. ; funding acquisition, S.H. and S.N. All authors have read and agreed to the published version of the manuscript.

**Funding:** This research received no external funding.

**Conflicts of Interest:** The authors declare no conflict of interest.

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
