# Peer review of "Analysis of Sustainable Public Transportation and Mobility Recommendations for Montevideo and Parque Rodó Neighborhood"

_smartcities, doi:10.3390/smartcities3020026_

Round 1
Reviewer 1 Report
I consider that your paper itself is well written and interesting to read, however I see the following major issues that should be resolved before publishing this paper:
The title of the article “analysis and recommendations for sustainable mobility in Montevideo and Parque Rodó Neighborhood” refers to “sustainable mobility, but the paper studies sustainable transportation initiatives recently developed in Montevideo. Even, the general recommendations to enhance sustainable mobility are related to transportation infrastructure or transportation systems, not to mobility. Urban and daily mobility studies are not just about transportation, they are about the people accessibility. Mobility is not synonymous with transportation, is more than that. Mobility studies analyse mobility at different levels and for different needs. My first recommendation is clarifying this distinction and adapt the title to “sustainable transportation”, what is the main topic of the paper. In this sense, the strength of this paper is the study of the initiatives of sustainable transportation of Montevideo regarding several quantitative and qualitative indicators. Hence, the focus of this paper should be the analysis of transportation initiatives of Montevideo.
Respect to introduction section, I recommend joining it with the second part dedicated to sustainable mobility. In the part dedicated to related work, it is exposed that Litman and Burwell “acknowledged the aforementioned issues for sustainable mobility” but the article of Litman and Burwell (2006) is about sustainable transportation. They establish relationships among sustainable transportation and mobility, but their main conclusion is dedicated to transport: “Most sustainable transport plans actually employ a combination of these approaches, including improved travel choices, pricing and road design incentives to encourage more efficient travel choices, land use patterns that reduce the need to travel and support alternative modes, and technical improvements to the motor vehicles that are used”.
Respect to “sustainable mobility initiatives in Montevideo” section, authors exhaustively describe and analyze sustainable transportation initiatives, but not sustainable mobility initiatives. The characteristics of public bicycles system; the pilot plan of electric bus; electric scooters; electric taxis and hybrid car sharing operating in Montevideo through public or private transportation companies are described. Some quantitative and qualitative transportation indicators are calculated (coverage for electric bus, public bicycles, and electric scooters services; affordability, computed for the three studied transportation modes considering short trips and long trips; middle and minimum income per capita in Montevideo and standard costs of every transport mode). Other characteristics, as comfort and pleasure, intermodal connectivity or energy efficiency are also briefly described. Although I consider this section is the strength of the research, it should be added a subsection dedicated to the promotion of walking as a very important sustainable transportation mode. I have also missed an analysis of the initiatives that are being implemented in Montevideo to limit or discourage the use of private motorized vehicle. What concerns to “suggestions and recommendations to develop and improve sustainable mobility in Montevideo”, they are ways to improve transportation system, planning or infrastructures.
The section named “Practical approach for analysis and implementation of a sustainable mobility plan for Engineering Faculty and Parque Rodó neighborhood, Montevideo” is presented as a “specific case study that demonstrate the viability of analyzing and implementing a sustainable mobility plan for Engineering Faculty and Parque Rodó neighborhood, Montevideo”. However, this part is not an analysis of viability. It is a descriptive analysis but it is not enough to demonstrate the feasibility of sustainable transportation modes considered in the paper. Results of the conducted survey should be presented in a different way to make the work more consistent. Firstly, Results of the survey related to mobility model of Montevideo may be part of a specific section dedicated to the characteristics of the mobility model in Montevideo. Can the authors offer results about mobility differences by age, gender, social class or other characteristics? It is well known that the mobility characteristics vary depending these variables. Sencondly, the conducted survey does not establish significant results of viability, a concept that requires economic or financial analysis. Anyway, an analysis of public acceptance can be developed in terms of public’s opinion and about sustainable transportation modes. In this sense, I recommend to build an index of social acceptance of sustainable transportation. It would be interesting knowing what the level of social acceptance of sustainable transportation is.
Definitely, I recommend change the focus to the analysis of sustainable public transportation initiatives in Montevideo, including walking, and the study of its public acceptance. A final question: Is it enough to implement sustainable transport without taking into account differential mobility needs? Maybe, the answer to this question could be a future line of research.
Finally, let me say that it was a pleasure to read this manuscript. I wish the authors of the best.
Reviewer 2 Report
I want to thank the authors for their interesting manuscript; there is some significant merit in this particular study and sufficient data, it appears, has been collected to warrant close consideration. However, I have some reservations about particular aspects of the research as it is currently explained and outlined, and some suggestions that need consideration and attention before I can recommend its publication.
The standard of English throughout needs to be improved to provide more coherence and clarity of thought.
A significant flaw in the manuscript is the failure to address the ‘elephant’ in the room, the automobile. The private car has dominated the urban landscape for decades and transport decision-makers in many cities, region and countries continue to adopt car-centric approaches and solutions to general issues of citizen mobility. The Sustainable Mobilities paradigm emerged as a direct response to this dominance of the car and is an attempt to reverse decisions and thinking that have been damaging to efforts to promote active and sustainable modes of transport so any research on improving or promoting public transport or active modes of travel must include a discussion on automobility.
With regards to sustainable mobility, I suggest a discussion on the merits (or otherwise) of Transport Oriented Development (TOD) is also warranted in such a text. The promotion of sustainable mobility options is doomed to failure without supportive urban planning and design, in my honest opinion
Figure 1 provided is somewhat surreptitious in the context of transport and mobility. Please clearly explain where you got this information from, why it is presented as it is depicted, and what it actually means in terms of sustainable mobility?
I suggest that section 4 (Sustainable mobility initiatives in Montevideo) ought begin at the top of page 6 where there authors write about project URU/17/G32.
I was somewhat confused as to the methodology. The manuscript indicates that the study of Parque Rodó neighbourhood was based on specific interviews from a sample of 617 persons moving to/from the studied area. Are these the same interviews used to firstly make Montevideo city wide analysis of sustainable mobility initiatives and, if so, how can this be justified? If these are separate interviews then this must be made much clearer, and some explanation as to the city-wide collection of data needs to be explicit. I also would like to know how the data was collected; was it online, face-to-face interviews, by phone, etc.
There were some good descriptive statistics provided in the latter half of the manuscript, and maybe more emphasis should be placed here. Furthermore, I would like to read the policy implications that flow from this study. For example, what should the municipality policy- and decision-makers do to better promote sustainable and active modes of travel in this district?
Reviewer 3 Report
- it would be worth expanding abstract with the more detailed information of the type of the analysis prepared,
- it would be worth expanding information about data collection during survey research,
- it would be worth to transform paragraph, which start from the words “This article extends the previous…”, to make it easier to read,
- in the line 229 and 248 project “FSDA_1_2018_1_154502” is used. Is it the name or acronym? It would be worth to delete that or expand,
- it would be worth expanding the paragraph 4.1 with describe state of the transport system in the Montevideo before sustainable mobility initiatives was started,
- it would be worth expanding the paragraph 4.1.1. by specific information about implementation of the electric buses e.g. type of the technology used for charging,
- it would be worth expanding the paragraph 4.3. The sources of the input data for the analysis should be described with more detail,
- it would be worth to change figure 2 and place all of the mobility initiatives on the one map with different layers,
- it would be worth expanding figure 3 by the legend - colour – population,
- it would be worth expanding information about zones in figure 3, because the zones of population haven’t the same size in whole map,
- it would be worth expanding information of the sample selection in the developed survey by information of choosing households in the analyse area,
- in the article there is lack connection between calculation in paragraph 4 with paragraph 5. It would be worth to expanding article by the calculation for the case study area and compare to the whole city,
The above-mentioned comments are purely editorial and do not affect my high opinion of the publication.
Round 2
Reviewer 1 Report
Dear editor and authors,
I consider that the manuscript has been significantly improved and now warrants publication in Smart Cities. Authors has improved the article taking into account all my comments as a reviewer.
I wish you all the best
Author Response
Thank you for all your comments and wishes.
Reviewer 2 Report
I am happy that the authors have addressed my concerns in this, their second draft, and this has greatly enhanced the manuscript and suggest publication after minor revision. I am still somewhat concerned with the quality of English which, at times, takes from the quality of the research. I again request that the authors fully proofread their manuscript for such minor grammatical errors (see some examples below). That stated, I am happy to leave these concerns with the editor for a final decision.
Lines 18-20
In some cases, travels demand long periods of time, disregarding the distance travelled, due to many reasons related to several mobility situations.
Line 29
For decades, automobile has dominated the urban landscape
Line 51-52
based on a survey performed to 617 citizens that travel from/to the studied area
Line 73-74
It became a crucial aspect for nowadays communities due to its direct implications on quality of human life
Author Response
Thank you for your suggestions. We did our best to check and correct all grammar through the article.